# The reaction of methyl peroxy and hydroxyl radicals as a major source of atmospheric methanol

Jean-François Müller[1], Zhen Liu[2,3], Vinh Son Nguyen[2], Trissevgeni Stavrakou[1], Jeremy N. Harvey[2] & Jozef Peeters[2]

Methyl peroxy, a key radical in tropospheric chemistry, was recently shown to react with the hydroxyl radical at an unexpectedly high rate. Here, the molecular reaction mechanisms are elucidated using high-level quantum chemical methodologies and statistical rate theory. Formation of activated methylhydrotrioxide, followed by dissociation into methoxy and hydroperoxy radicals, is found to be the main reaction pathway, whereas methylhydrotrioxide stabilization and methanol formation (from activated and stabilized methylhydrotrioxide) are viable minor channels. Criegee intermediate formation is found to be negligible. Given the theoretical uncertainties, useful constraints on the yields are provided by atmospheric methanol measurements. Using a global chemistry-transport model, we show that the only explanation for the high observed methanol abundances over remote oceans is the title reaction with an overall methanol yield of ∼30%, consistent with the theoretical estimates given their uncertainties. This makes the title reaction a major methanol source (115 Tg per year), comparable to global terrestrial emissions.

[1] Atmospheric Composition Department, Royal Belgian Institute for Space Aeronomy, Avenue Circulaire 3, B-1180 Brussels, Belgium. [2] Department of Chemistry, University of Leuven, B-3001 Heverlee, Belgium. [3] State Key Laboratory of Chemical Engineering, East China University of Science and Technology, Meilong Road 130, Shanghai 200237, China. Correspondence and requests for materials should be addressed to J.-F.M. (email: jfm@aeronomie.be) or to J.P. (email: jozef.peeters@kuleuven.be).

Methyl peroxy radical ($CH_3O_2$) is the most important organic peroxy radical in the atmosphere, with a global production (~50 Tmoles or 2,500 Tg per year) primarily due to the oxidation of ubiquitous methane by hydroxyl (OH) radicals[1]. Its reaction with nitric oxide (NO) accounts for ~25% of the global production of tropospheric ozone[2,3], second in importance only to $HO_2 + NO$. Under pristine conditions, $CH_3O_2$ reacts chiefly with hydroperoxy radicals $HO_2$ to form methyl hydroperoxide $CH_3OOH$, depleting odd hydrogen radicals ($HOx \equiv OH + HO_2$) directly and through the further reaction of $CH_3OOH$ with OH. The reaction of $CH_3O_2$ with organic peroxy radicals ($RO_2$) is only a minor sink[4], but it is also the only well-documented photochemical source of atmospheric methanol, amounting to 18–38 Tg per year globally according to model estimates[5]. Note that a much larger total photochemical source (50–100 Tg per year) was invoked by Jacob et al.[4] to rationalize serious model underestimations of methanol observations during an aircraft campaign over the Pacific[6].

Although the possibility of a reaction of $CH_3O_2$ with OH was previously considered[7,8], it is only recently that a direct, absolute determination of its rate has been reported. Coupling Laser Induced Fluorescence and cw-Cavity Ring-Down Spectroscopy to laser photolysis, Bossolasco et al.[9] measured a rate of $(2.8 \pm 1.4) \times 10^{-10}$ cm$^3$ molecule$^{-1}$ s$^{-1}$. This unexpectedly high value makes this reaction a major sink of $CH_3O_2$ in pristine conditions. Using campaign data in Cape Verde, Fittschen et al.[10] estimated that it accounts for ~25% of the overall $CH_3O_2$ sink at that remote site, rivalling the reaction with $HO_2$. Its impact on atmospheric chemistry is however critically dependent on the nature and yields of the products. Three exothermic overall reaction channels were envisaged[8]:

$$OH + CH_3O_2 \rightarrow CH_2O_2 + H_2O \qquad (1)$$

$$OH + CH_3O_2 \rightarrow CH_3O + HO_2 \qquad (2)$$

$$OH + CH_3O_2 \rightarrow CH_3OH + O_2 \qquad (3)$$

with $CH_2O_2$ the singlet Criegee intermediate, $H_2C=O^{(+)}-O^{(-)}$. The Criegee channel (1) has been invoked as the possible source of a missing oxidant of $SO_2$ inferred from observations at a coastal site[11]. A large Criegee yield would also lead to a very large source of formic acid through reaction of stabilized $CH_2O_2$ with water[12]. The reaction was therefore speculated[8,13] to explain part of the missing source of HCOOH required to sustain the high HCOOH levels observed in the atmosphere[14,15], although it was found to degrade model/data correlation for measurement campaigns over the U.S.[13]. The methoxy channel (2) leads ultimately to two $HO_2$ radicals and formaldehyde, which is the end product of both $CH_3O_2 + NO$ and (through $CH_3OOH$) $CH_3O_2 + HO_2$. The methanol channel (2) was noted to be a potentially very significant source of methanol[8,13], but no quantitative assessment has been made to this date.

A recent quantum chemical study of $CH_3O_2 + OH$ by Bian et al.[16] could identify only one thermally accessible pathway commensurate with the high measured rate constant: combination of the reactants into an activated methylhydrotrioxide $CH_3OOOH$ (**TRIOX**), followed by direct dissociation into $CH_3O$ and $HO_2$, which are about 4 kcal mol$^{-1}$ more stable than the reactant radicals. Another recent theoretical study by Nguyen et al.[17] briefly addressed the reaction, concluding that the dominant pathway in atmospheric conditions is collisional stabilization of $CH_3OOOH$, whereas production of Criegee is negligible.

In view of the likely major importance of the title reaction for key oxygenated organic compounds and its potentially large impact on HOx radicals in the remote troposphere, this work proposes to: (i) elucidate the molecular mechanisms of the reaction by constructing detailed potential energy surfaces, using suitable high-level density functional theory (DFT) and ab initio methodologies for the singlet and triplet (biradical) intermediates; (ii) identify the kinetically viable reaction channels and distinguish between the major and minor product routes using appropriate statistical rate theories; (iii) use a global chemistry-transport model, the Intermediate Model for the Annual and Global Evolution of Species (IMAGES, see the 'Methods section'), to assess the impact of the reaction and constrain the yields through comparisons with atmospheric measurements. For those readers most interested in the implications for atmospheric chemistry, the results relating to goals (i) and (ii) are succinctly summed up before the section on atmospheric modelling, and we invite such readers to move directly to this summary.

## Results

**Potential energy surface and reaction kinetics.** We characterized all three exothermic overall reaction channels (reactions (1), (2), (3)) and all other relevant pathways of the title reaction using quantum chemical methods. Structures and vibrational frequencies were computed with DFT, at the M06-2X-D3/6-311++ G(3df,3pd) level of theory (Supplementary Fig. 1, Supplementary Fig. 2, and Supplementary Table 1). More accurate energies were computed using coupled-cluster theory, with explicit ('F12') treatment of electron–electron distances, as shown in Fig. 1. The DFT and coupled-cluster calculated properties of various complexes and transition states are summarized in Table 1.

The reaction of $CH_3OO$ with OH starts by the barrier-less formation of singlet reactant complex **$^1$RC** and triplet reactant complex **$^3$RC**, in which OH donates a hydrogen bond to $CH_3OO$, and in which the singlet and triplet states are near degenerate. Hydrogen abstraction starting from **$^1$RC** and **$^3$RC** to form the Criegee intermediate $^1CH_2OO$ and the triplet biradical $^3CH_2OO$ via **TS2** and **TS3**, respectively, involve significant barriers. In absence of intersystem crossing (ISC) **$^3$RC→$^1$RC**, the reaction $CH_3O_2 + OH \leftrightarrow {}^3RC$ will be quasi-equilibrated such that the bimolecular rate constant for the overall triplet channel $CH_3O_2 + OH \rightarrow {}^3CH_2O_2 + H_2O$ can be found from transition state theory (TST)[18–20], yielding $^3k_{bi,3}(298\,K) = 2.6 \times 10^{-15}$ cm$^3$ molecule$^{-1}$ s$^{-1}$. This channel is therefore entirely negligible, as are the other triplet entrance routes described by Bian et al.[16], all proceeding through transition states that lie even much higher. Instead, rapid ISC of **$^3$RC** to the nearly-degenerate **$^1$RC** could channel much of the triplet entrance flux also towards the singlet surface and hence to **TRIOX** (see below), rationalizing the high measured overall rate constant[9] of $2.8 \times 10^{-10}$ cm$^3$ molecule$^{-1}$ s$^{-1}$.

The favoured reaction channel of the chemically activated **$^1$RC** is the formation of the stable methylhydrotrioxide **TRIOX** over the very low-energy **TS1**. Direct dissociation of **TRIOX** into $CH_3O$ and $HO_2$ is not a minimum energy pathway; instead, the energetically favoured decomposition route passes through the singlet product complex **$^1$PC** over **TS4**, which involves concerted $CH_3O$–OOH bond-breaking and formation of a $CH_3O$–HOO hydrogen bond. The relative energy and vibrational properties of saddle-point **TS4** play an important role in determining selectivity, but this point is also very difficult to characterize accurately. As explained below, the best results are obtained by using a structure optimized with the M06-D3 DFT method, rather than M06-2X-D3 as used for the other stationary points. Like the reactant complex, the hydrogen-bonded **PC** has near-degenerate singlet and triplet states. From **$^1$PC**, $CH_2O + HOOH$ can be formed through **TS5**, or $CH_3OH + {}^1O_2$ can be obtained through **TS7**. Equally, **$^1$PC** can convert to triplet

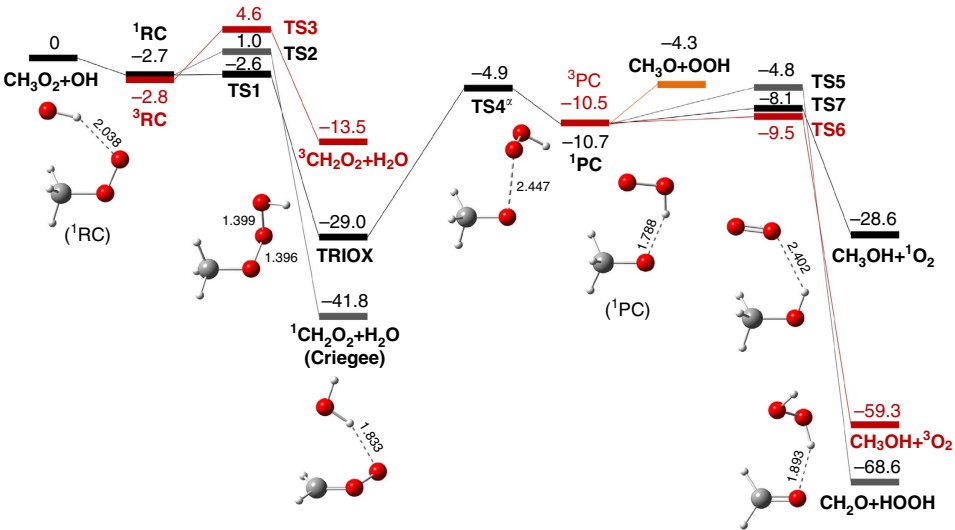

**Figure 1 | Potential energy surface for CH₃OO + OH biradical reactions.** Coupled-cluster energies (kcal mol⁻¹) relative to separated reactants $CH_3O_2$ and OH, based on DFT geometries (CCSD(T)-F12/cc-pVTZ-F12//M06-2X-D3/6-311++G(3df,3pd) level of theory). TS, transition state; RC, reactant complex; PC, product complex. The singlet reaction pathways are depicted in black and grey, while the triplet reaction pathways are depicted in red for clarity. ᵃTS4 was optimized at M06-D3/6-311++G(3df,3pd) level of theory, see text (each structure with formula or acronym is depicted separately in Supplementary Fig. 2).

**Table 1 | Relative energies with inclusion of ZPVE for all structures involved in the title reaction.**

| Complexes | $\Delta E^\star$ | $\Delta E^\dagger$ | T1 diag.‡ |
|---|---|---|---|
| $CH_3O_2 + OH$ | 0 | 0 | – |
| ¹RC | − 4.4 | − 2.7 | 0.025 |
| ³RC | − 4.5 | − 2.8 | 0.024 |
| TRIOX | − 29.9 | − 29.0 | 0.015 |
| ¹PC | − 12.3 | − 10.7 | 0.028 |
| ³PC | − 12.1 | − 10.5 | 0.025 |
| TS1 | − 4.3 | − 2.6 | 0.026 |
| TS2 | − 0.1 | 1.0 | 0.035 |
| TS3 | 3.0 | 4.6 | 0.028 |
| TS4 | − 7.1 | − 4.9 | 0.027 |
| TS5 | − 6.3 | − 4.8 | 0.038 |
| TS6 | − 12.0 | − 9.5 | 0.044 |
| TS7 | − 10.9 | − 8.1 | 0.081 |
| $CH_3O + HO_2$ | − 4.8 | − 4.3 | – |
| $CH_3OH + {}^3O_2$ | − 60.1 | − 59.3 | – |
| $CH_2O + H_2O_2$ | − 67.4 | − 68.6 | – |

*Computed at M06-2X-D3/6-311++G(3df, 3pd) level of theory.
†Computed at CCSD(T)-F12/cc-pVTZ-F12//M06-2X-D3/6-311++G(3df, 3pd) level of theory.
‡T1 diagnostic for CCSD(T)-F12 calculations.

product complex **³PC**, and **³PC** can form $CH_3OH + {}^3O_2$ through the very low **TS6**.

Transition states **TS2**, **TS3**, **TS5**, **TS6** and **TS7** corresponding to the abstraction or transfer of a hydrogen atom are straightforward and their identity has been carefully confirmed using intrinsic reaction coordinate (IRC) calculations. The coupled-cluster relative energies agree well with those from DFT, supporting the use of the M06-2X-D3 functional for geometry optimization. However, **TS1** and **TS4** are more problematic and need further discussion, in particular as these crucial transition states (TSs) largely determine the dominant reactant pathways. Both TSs can be considered to correspond to breaking of one of the O–O bonds in **TRIOX**. However, this step does not proceed with monotonously increasing energy towards the fragments, due to the existence of the hydrogen-bonded **RC**

and **PC**. As the relative orientation of the fragments in the complexes differs from that in **TRIOX**, the TSs have combined O–O bond breaking and fragment reorientation character. Also, **TRIOX** has a closed-shell singlet electronic structure, with the transition to the open-shell radical pair nature of **PC** or **RC** occurring in the region near the TSs. For open-shell singlets with completely uncoupled electrons, singlet and triplet states are near-degenerate (as in **RC** and **PC**). Unrestricted DFT calculations and coupled-cluster calculations based on unrestricted Hartree-Fock reference wavefunctions are reasonably accurate in this case. However, for partially uncoupled electrons, the triplet is significantly higher in energy, and unrestricted approaches are less accurate, introducing artefacts in the shape of the energy curve. For **TS1**, we are able to locate *two* saddle points, **TS1** and **TS1′**, with the M06-2X-D3 functional. The first one corresponds mostly to reorientation of the OH fragment, while **TS1′** has essentially only O–O bond stretching character. Large-active-space CASSCF (complete active space self-consistent field) and CASPT2 (complete active space perturbation theory 2) calculations (Supplementary Fig. 3 and Supplementary Note 1), which can treat systems with partially uncoupled electrons in a balanced way, strongly suggest **TS1′** to be purely an artefact based on inaccurate description of the developing low-spin open-shell electronic structure, whereas the **TS1** is more reliable.

For **TS4**, only one TS structure, **TS4′**, is obtained with M06-2X-D3, but CASSCF and CASPT2 calculations show it is an artefact, similar to **TS1′**. In contrast, the M06-D3 functional yields a TS structure, denoted **TS4**, which is closer to the correct TS according to CASSCF and CASPT2. The difference in structure is quite large: the M06-2X-D3 **TS4′** structure is very 'early' (O–O distance of 1.967 Å) whereas the M06-D3 **TS4** has an O–O distance of 2.447 Å. Accordingly, we have used the **TS4** structure in this study. For the other stationary points, the CCSD(T)-F12 (coupled cluster—single, double and triple excitation theory) total energies at the M06-D3 and M06-2X-D3 structures differ by <0.2 kcal mol⁻¹, and the vibrational frequencies are very similar also.

The chemically activated singlet reactant complex **¹RC**, with average vibrational energy $<E_v> = 4.8$ kcal mol⁻¹, including an average 2.1 kcal mol⁻¹ thermal energy at 298 K inherited from

the reactants (see the 'Methods section'), reacts nearly barrier-free (**TS1**) to form the closed-shell $CH_3OOOH$ molecule, **TRIOX**, much faster than forming Criegee $^1CH_2O_2$ and $H_2O$ over a barrier of 3.7 kcal mol$^{-1}$ (**TS2**). The RRKM-based (Rice, Ramsperger, Kassel, Marcus)[21,22] unimolecular rate coefficients averaged over the (narrow) distribution function of formation $F(E_{th,v})$ (see the 'Methods section') are $<k_1> = 1.8 \times 10^{13}$ s$^{-1}$ and $<k_2> = 3.4 \times 10^{10}$ s$^{-1}$, respectively. The $k$-subscripts refer to the TS numbering in Fig. 1. All $F(E_{th,v})$-averaged rate coefficients of the activated reaction steps given in this subsection are for 298 K and 1,013 hPa (air). They are listed also in Table 2, together with the values for 285 K and 750 hPa, as well as these for 256 K and 400 hPa (see next subsection). Since the other pathways to $^1CH_2O_2 + H_2O$ theoretically characterized by Bian *et al.*[16] and Nguyen *et al.*[17] contribute even less as they involve TSs lying $\geq 7$ kcal mol$^{-1}$ above the initial reactants, it is clear that Criegee formation contributes not more than $\sim 1\%$ overall.

**TRIOX**, with average chemical activation energy $<E_v>$ of 31.1 kcal mol$^{-1}$ (at 298 K) converts rapidly to the product complex $^1$**PC** over the low-lying and very loose **TS4**, characterized above, far outrunning the other isomerization/decomposition reactions through high-lying transition states[16]. The rate coefficient integrated over the formation distribution function $F(E_{th,v})$ is evaluated at $<k_4> = 2.4 \times 10^{10}$ s$^{-1}$, implying a **TRIOX** lifetime of $\sim 40$ ps during which it should lose $\sim 0.45$ kcal mol$^{-1}$ by collisions at 298 K and 1,013 hPa (see below), shifting down also the energy distribution $F(E_{th,v})$ for $^1$**PC**. The $CH_3O \bullet HO_2$ complex $^1$**PC** has a decisive role in our system; the $\sim 7$ kcal mol$^{-1}$ strong $CH_3O$–HOO hydrogen bond explains why the minimum energy pathway of $CH_3OOOH$ to $CH_3O + HO_2$ passes through it. The fastest reaction of $^1$**PC**, with $<E_v>(^1$**PC**$) \approx 12.4$ kcal mol$^{-1}$, is dissociation into these radicals by breaking the H-bond without exit barrier. The $F(E_{th,v})$- averaged $<k_{diss}>$ was estimated by variational RRKM[21,22]. The energy, zero-point vibrational energy (ZPVE) and integrated density of states $G^{var}(E_v - E^{var})$ were computed for a series of structures with increasing H-bond length, using the unprojected vibrational frequencies, as listed in Supplementary Table 2; the use of projected frequencies[23–25] for the sum of states would have resulted in $\sim 12\%$ higher $<k_{diss}>$ and $<^3k_{diss}>$, with a minor effect on the overall product yields (see Supplementary Note 3). The variational bottleneck that minimizes $G^{var}(E_v - E^{var})$, shown in Fig. 2, was found for an H-bond length of 3.2 Å, and relative

energy 1.7 kcal mol$^{-1}$ below the separated $CH_3O + HO_2$, giving $<k_{diss}> \approx 2.4 \times 10^{12}$ s$^{-1}$.

The decomposition of $^1$**PC** into $CH_2O + H_2O_2$ over the fairly high **TS5**, with RRKM-calculated rate of only $<k_5> = 9.7 \times 10^9$ s$^{-1}$ is negligibly slow, whereas decomposition into $CH_3OH + O_2(^1\Delta)$ over **TS7**, at averaged rate $<k_7> = 1.2 \times 10^{11}$ s$^{-1}$, is a channel of atmospheric relevance. Importantly, conversion of the initially formed $^1$**PC** to $^3$**PC** will affect the relative yields of the different products. Spin–orbit coupling between the lowest singlet and triplet states in **PC** is very small, but the spin–orbit coupling between the lowest singlet state and the second-lowest triplet state (or between the second-lowest singlet and the lowest triplet) is much larger, as there is now an orbital angular momentum difference, and we calculate a root-mean-square coupling matrix element of 58 cm$^{-1}$ in both cases. Simply assuming Rabi cycling between the two states, a 'rate constant' for singlet-triplet conversion of $1.7 \times 10^{12}$ s$^{-1}$ is estimated. Using a more rigorous statistical rate theory for reactions with spin-state change[26,27] (see Supplementary Note 2), we obtain a remarkably similar estimate of $<k_{ISC}> = 3.5 \times 10^{12}$ s$^{-1}$, while the reverse $<k_{-ISC}> = 1.9 \times 10^{12}$ s$^{-1}$. The reverse reaction $^1$**PC**→**TRIOX** occurs with an average rate $<k_{-4}>$ of $1.3 \times 10^{11}$ s$^{-1}$, which is $\sim 3\%$ of the total $^1$**PC** removal rate, such that the net **TRIOX**→$^1$**PC** conversion rate $k_4^n(E_v) = 0.97 \times k_4(E_v)$.

$^3$**PC** decomposes into $CH_3OH$ and ground state $^3O_2$ over a very low barrier (**TS6**), with RRKM-calculated average $<^3k_6>$ of $4.1 \times 10^{11}$ s$^{-1}$, faster than the singlet $^1$**PC** reaction to $CH_3OH$,

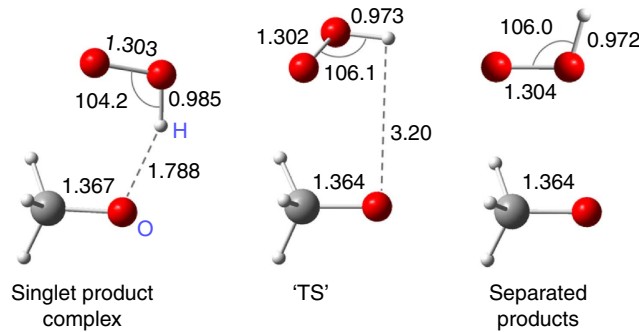

**Figure 2 | Dissociation of product complex and the variational transition state.** Bond lengths are in angstrom. Angles are in degrees.

**Table 2 | RRKM-calculated rate coefficients $<k(E_v)>$ averaged over the thermal energy distribution of formation $F(E_{th,v})$\* for the various chemically activated reactions.**

| Reaction step | Notation | $k$(s$^{-1}$) (298 K) (1,013 hPa) | $k$(s$^{-1}$) (285 K) (750 hPa) | $k$(s$^{-1}$) (256 K) (400 hPa) |
|---|---|---|---|---|
| $^1$RC → TRIOX | $<k_1>$ | $1.82 \times 10^{13}$ | $1.81 \times 10^{13}$ | $1.76 \times 10^{13}$ |
| $^1$RC → $^1CH_2O_2 + H_2O$ | $<k_2>$ | $3.36 \times 10^{10}$ | $2.56 \times 10^{10}$ | $1.20 \times 10^{10}$ |
| TRIOX → $^1$PC | $<k_4>$ | $2.38 \times 10^{10}$ | $2.25 \times 10^{10}$ | $1.90 \times 10^{10}$ |
| $^1$PC → TRIOX | $<k_{-4}>$ | $1.29 \times 10^{11}$ | $1.29 \times 10^{11}$ | $1.23 \times 10^{11}$ |
| $^1$PC → $CH_3O + HO_2$ | $<k_{diss}>$† | $2.43 \times 10^{12}$ | $2.41 \times 10^{12}$ | $2.32 \times 10^{12}$ |
| $^1$PC → $^3$PC | $<k_{isc}>$ | $3.50 \times 10^{12}$ | $3.50 \times 10^{12}$ | $3.44 \times 10^{12}$ |
| $^3$PC → $^1$PC | $<^3k_{-isc}>$‡ | $1.87 \times 10^{12}$ | $1.87 \times 10^{12}$ | $1.84 \times 10^{12}$ |
| $^1$PC → $CH_2O + H_2O_2$ | $<k_5>$ | $9.67 \times 10^9$ | $9.67 \times 10^9$ | $9.29 \times 10^9$ |
| $^3$PC → $CH_3O + HO_2$ | $<^3k_{diss}>$†,‡ | $3.37 \times 10^{12}$ | $3.36 \times 10^{12}$ | $3.23 \times 10^{12}$ |
| $^3$PC → $CH_3OH + ^3O_2$ | $<^3k_6>$‡ | $4.14 \times 10^{11}$ | $4.14 \times 10^{11}$ | $4.14 \times 10^{11}$ |
| $^1$PC → $CH_3OH + ^1O_2$ | $<k_7>$ | $1.20 \times 10^{11}$ | $1.20 \times 10^{11}$ | $1.20 \times 10^{11}$ |

\*For formation of $^1$**RC** and **TRIOX**, $F(E_{th,v})$ is the initial thermal distribution; for $^1$**PC** and $^3$**PC**, $F(E_{th,v})$ is shifted down by collisions, by 0.45, 0.35 and 0.25 kcal mol$^{-1}$ at 298, 285 and 256 K, respectively (see text).
†$k$-value obtained using variational RRKM.
‡Rate coefficient of triplet $^3$**PC** reaction preceded by superscript 3 for clarity.

above. It must compete with the reverse ISC, above and also with the fast dissociation of $^3$**PC** into $CH_3O + HO_2$ at variational-RRKM rate $<^3k_{diss}> \approx 3.4 \times 10^{12}\,s^{-1}$ (see data in Supplementary Table 3), but still results in twice more $CH_3OH + {^3}O_2$ production than the $CH_3OH + {^1}O_2$ afforded by $^1$**PC**, above. Accounting for the rates of all reactions of $^1$**PC** and $^3$**PC**, the two latter routes together are found to result in a $CH_3OH$ yield through activated **TRIOX** of 7.8% at 298 K and 1,013 hPa. Given the possible error of $\sim 1.5\,kcal\,mol^{-1}$ on the energies of the transition states to $CH_3OH$ relative to the (variational) transition states of the **PC** dissociations, and taking into account also a likely error of a factor of $\sim 2$ on the ratio of $k_{ISC}/k_{diss}$, we estimate an uncertainty margin on the $CH_3OH$ yield of a factor of $\sim 3.5$. Note that about 45% of the products of activated **TRIOX** arise via ISC of $^1$**PC** to $^3$**PC**. Of major importance is that **TRIOX** does not decompose directly but via the complex $^1$**PC**, which enables production of $CH_3OH$, while direct methanol formation from $CH_3OOOH$ would face a quasi-unsurmountable barrier[16].

A (minor) fraction of the activated **TRIOX** will suffer energy loss by successive collisions with air molecules to yield thermalized $CH_3OOOH$, of which the subsequent fates are discussed in Supplementary Note 4. Using the bi-exponential energy transfer model of Troe[28] with an assumed average energy transferred per collision $<\Delta E>_{all}$ of $-0.9\,kcal\,mol^{-1}$—fairly

high because **TRIOX** has several low-frequency vibration modes—and taking into account the distribution function $F(E_{th,v})$ of the activated **TRIOX**, the fraction of stabilization at a collision frequency of $1.2 \times 10^{10}\,s^{-1}$ at 298 K and 1,013 hPa was evaluated (see the 'Methods section') to be $f_{stab} \approx 10.7\%$. However, this result is quite uncertain, first of all because it is very sensitive to the assumed value of $<\Delta E>_{all}$: doubling it increases $f_{stab}$ nearly threefold while halving it reduces $f_{stab}$ to $< 2\%$. Moreover, the calculated $f_{stab}$ depends also strongly on $k_4(E_v)$, which itself bears a possible error of a factor $\sim 1.5$–2. We therefore estimate a stabilized $CH_3OOOH$ yield at 298 K and 1 atm in the range 0–35%. The much higher $f_{stab}$ of $\sim 90\%$ predicted by Nguyen et al.[17] can be ascribed to their TS for **TRIOX**$\rightarrow{^1}$**PC** conversion being our artefactual **TS4'**, which on account of its far higher rigidity than **TS4** leads to a much lower calculated rate of **TRIOX**$\rightarrow{^1}$**PC** and hence much more **TRIOX** stabilization.

**Reaction products and estimated yields in the troposphere.** The theoretical investigation above predicts that the dominant product route of the title reaction is channel (2) yielding the radicals $CH_3O + HO_2$, whereas the suggested channel (1)[8,9] producing the Criegee Intermediate $CH_2O_2$ is found to be entirely negligible. On the other hand, we find that the title reaction yields a sizable fraction of collisionally thermalized (but chemically labile) $CH_3OOOH$, of order of 10% at 1 atm and 298 K, while we also uncovered two parallel pathways leading directly to $CH_3OH + O_2$ with overall yield around 7% in the same conditions. No evidence could be found for any other significant product routes. It must be stressed that the yield estimates of $CH_3OOOH$ and $CH_3OH$ given above and listed in Table 3 are subject to large uncertainty factors of $\sim 3.5$ for each, as detailed in the previous subsection.

As detailed in the next subsection, the title reaction is most important above the tropical oceans, where NO levels are low but OH concentrations moderate to high. The rate of the reaction as a function of decreasing pressure (or of increasing altitude) and as a function of latitude is depicted in Supplementary Fig. 4. Near the Equator, accounting for the temperature profile, $\sim 50\%$ of the

**Table 3 | Predicted products of the $CH_3O_2 + OH$ reaction and best-estimate yields at various pressures and corresponding average temperatures in the troposphere above the tropics.**

| Reaction products | Reaction channel | 1,013 hPa, 298 K | 750 hPa, 285 K | 400 hPa, 256 K |
|---|---|---|---|---|
| $CH_3O + HO_2$ | (2) | 0.82 | 0.85 | 0.88 |
| $CH_3OOOH$* | (4)* | 0.107 | 0.074 | 0.035 |
| $CH_3OH + O_2$† | (3) | 0.069 | 0.072 | 0.078 |

*Channel 4: $CH_3O_2 + OH \rightarrow$ thermalized $CH_3OOOH$.
†Overall yield of $CH_3OH$ formed through activated $CH_3OOOH$†.

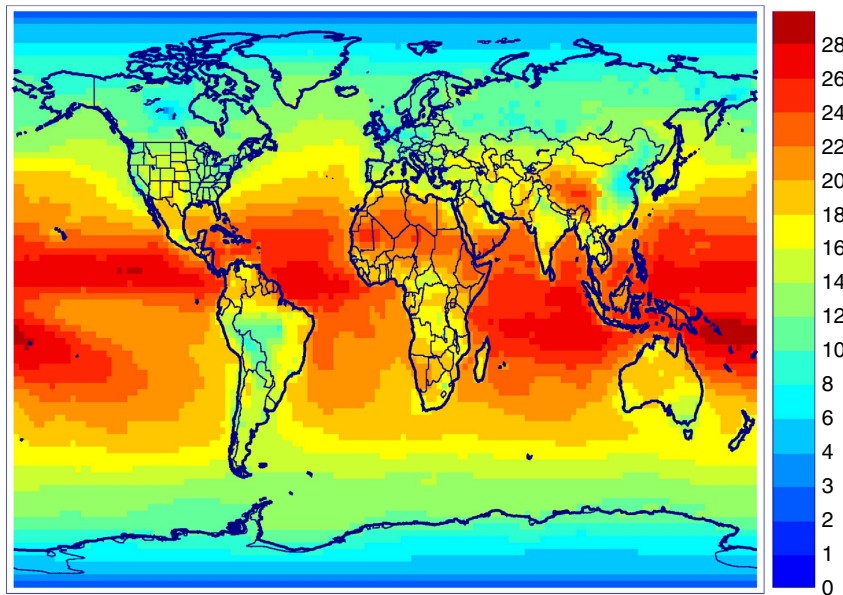

**Figure 3 | Contribution of $CH_3O_2 + OH$ to the sink of $CH_3O_2$.** Modelled yearly averaged contribution (%) of the reaction to the vertically integrated sink of $CH_3O_2$. Map created using IDL version 8.2.3.

reaction occurs below 750 hPa (altitude $\sim 2.4$ km) where the average temperature is 285 K, and nearly $\sim 95\%$ below 400 hPa (altitude $\sim 7.5$ km) where temperature is 256 K. Given that the $CH_3OOOH$ stabilization fraction should show a considerable pressure and temperature dependence, and a minor temperature dependence is also expected for other pathways (see the 'Methods section') we have evaluated the $CH_3OOOH$ and $CH_3OH$ yields also for the 'median' conditions 750 hPa and 285 K, and for the 'limit' conditions 400 hPa and 256 K, above. The results, together with those for 1,013 hPa and 298 K, are listed in Table 3. The pronounced pressure- and temperature dependences of the $CH_3OOOH$ stabilization fraction can be expressed within a few per cent by the power function $f_{stab}(P,T) = 0.107 \times (P/1,013 \text{ hPa})^{2.0} \times (298 \text{ K}/T)^{5.0}$.

**Modelled global impact.** Adopting the measured rate[9] of $CH_3O_2 + OH$, the reaction accounts for 20% of the global $CH_3O_2$ sink according to IMAGES simulations. The global flux through that reaction is considerable: $\sim 11$ Tmoles per year, comparable to, for example, the global source of isoprene[29]. Although it is only a minor sink ($\sim 10\%$) of $CH_3O_2$ at high latitudes and over polluted continental regions (Fig. 3) where the reaction with NO is by far the largest sink (Supplementary Fig. 5), the reaction with OH represents 20–30% of the sink over tropical oceans and deserts, where $CH_3O_2 + HO_2$ is however still dominant (30–50% of total sink).

Table 4 summarizes the model simulations. The best theoretical estimates are used in run B, while runs C and D adopt yields approaching respectively the high end and low end of the estimated uncertainty range of both the direct $CH_3OH$ yield and the **TRIOX** stabilization fraction. Relative to a simulation neglecting the title reaction, the $HO_2$ radical abundances are increased by 10–20% over tropical oceans (Supplementary Fig. 6). Unsurprisingly, the impact is highest when the $HO_2$ yield is highest (run D). The changes are negligible or even negative in high-NOx areas.

The changes in OH are of opposite sign to those of $HO_2$, reaching up to $-7\%$ over remote oceans. Loss of OH due to reaction with $CH_3O_2$ and the products $CH_3OOH$ and $CH_3OH$ is partially offset by OH recycled from enhanced $HO_2$. The globally averaged OH concentration decreases by 1.5–3.2% depending on the simulation, increasing the methane lifetime by up to 0.3 years in run C. The largest OH changes are calculated in run C, due to its lower $HO_2$ yield and higher OH loss through reaction with $CH_3OOOH$ and $CH_3OH$. Surface ozone is decreased due to the $CH_3O_2 + OH$ reaction, by up to 6% over remote oceans, and by 1–2% ($< 1$ ppbv) over Europe, North America and East Asia in July (Supplementary Fig. 7). Hydrogen peroxide ($H_2O_2$)

is strongly impacted, owing to the quadratic dependence of its production on $HO_2$ levels. Its concentrations increase by up to 30% in run C (Fig. 4) and $\sim 50\%$ in runs B and D.

Methyl peroxy radical abundances are reduced by up to 40% over tropical oceans (Fig. 4), irrespective of the yield assumption. This strong drop in concentration reflects increased loss through reaction with OH and $HO_2$, and slightly decreased production from $CH_4 + OH$. The decreases in $CH_3OOH$ are also substantial (up to $-30\%$). Formaldehyde is almost unaffected, being an end product to all $CH_3O_2$ sink pathways. Methanol, however, shows drastic differences among the simulations: whereas its concentrations increase or decrease by $\sim 20\%$ at most in runs B and D, the global methanol burden is increased by 60% in run C, with concentration increases reaching 30–100% over remote oceans (Fig. 4). Even over continents, methanol increases by 10–20% over most areas and by 80–200% over desert regions with very low methanol emissions. Methanol formation from $CH_3O_2 + OH$ is largely due to the direct pathway to $CH_3OH + O_2$ (56%), but indirect formation through stabilized **TRIOX** is significant (44%). Reaction on aerosols is calculated to be the largest **TRIOX** sink globally, followed by reaction with OH and reaction with $(H_2O)_2$ (Supplementary Fig. 8, Supplementary Note 4). The calculated average yield of methanol from stabilized **TRIOX** is 65%.

**Model evaluation for peroxides and formaldehyde.** We limit our evaluation to stable compounds over oceans, where strongest impacts are expected. Both the modelled vertical profile and latitudinal profile of $H_2O_2$ agree generally fairly well with the observations from aircraft and ship campaigns (Supplementary Fig. 9, Supplementary Fig. 10, and Supplementary Fig. 11), indicating that its sources and sinks are reasonably well described by the model. As seen in Supplementary Table 4, the average model bias across all campaigns is improved when including the title reaction, from $-14\%$ in run A to $+3$ and $-2\%$ in runs B and C, respectively. Although this improvement could be fortuitous, given the known uncertainties in $HO_x$ modelling[30], the title reaction clearly does not lead to noticeable inconsistencies with the data. As expected, the reaction has negligible impact on modelled $CH_2O$ (Supplementary Fig. 12) which agrees very well with aircraft data over oceans.

For $CH_3OOH$, the title reaction leads to model underestimations ($> \sim 25\%$) of airborne measurements, but it improves significantly the model agreement with ship measurements (Supplementary Fig. 11, Supplementary Fig. 13, and Supplementary Table 5). The contradiction between the conclusions from either ship or aircraft data is difficult to explain given the wide geographical area covered by both platforms. As in a previous modelling study[13], the largest biases are

**Table 4 | Overview of model simulations with assumed product yields.**

| Run | $CH_3O_2 + OH$ | $CH_3O$ | TRIOX | $CH_3OH$ | $CH_2OO$ |
|---|---|---|---|---|---|
| A | Ignored | – | – | – | – |
| B | Best estimate | 0.86 | 0.07 | 0.07 | 0 |
| C | High methanol case* | 0.61 | 0.21 | 0.18 | 0 |
| D | Low methanol case* | 0.975 | 0 | 0.025 | 0 |
| E | High Criegee case† | 0.6 | 0 | 0 | 0.4 |
| A_NO | As A, no ocean source‡ | – | – | – | – |
| C_NO | As C, no ocean source‡ | 0.61 | 0.21 | 0.18 | 0 |
| C_VR | As C, low k($CH_3OOH + OH$)§ | 0.61 | 0.21 | 0.18 | 0 |

*Methanol yield from activated trioxide multiplied (divided) by 3 in run C (D) relative to best estimate. Stabilisation fraction multiplied by 3 in run C, taken equal to zero in run D.
†Not a theoretical prediction.
‡Oceanic methanol emission omitted.
§Use lower rate for reaction $CH_3OOH + OH$, within recommended uncertainty range.
Globally averaged molar yields of $CH_3O$, stabilized trioxide, $CH_3OH$ and $CH_2OO$ adopted in model runs. The yields are pressure- and temperature-dependent (see text).

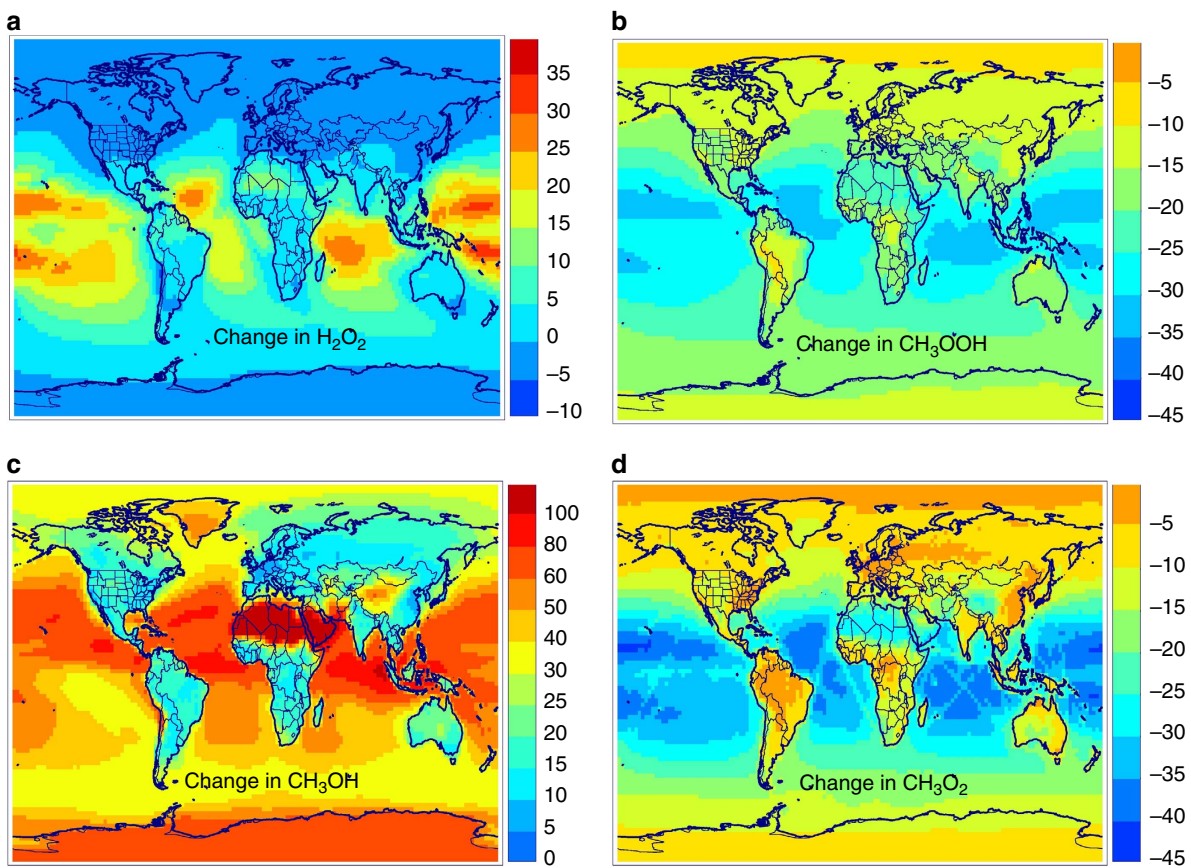

**Figure 4 | Impact of $CH_3O_2 + OH$ on key compounds abundances.** Annually averaged modelled changes (in %) in near-surface concentrations of (**a**) $H_2O_2$, (**b**) $CH_3OOH$, (**c**) $CH_3OH$ and (**d**) $CH_3O_2$ in the high methanol simulation (run C). Maps created using IDL version 8.2.3.

found for INTEX-B (ref. 31). Since even the run ignoring the title reaction largely underestimates the observations in several campaigns, measurement issues and/or model uncertainties likely cause the discrepancies. For example, the estimated uncertainty in the rate constant of the $CH_3OOH + OH$ reaction is a factor of 1.4 (ref. 32). Adopting a rate constant measurement[33] about 25% lower than the current Jet Propulsion Laboratory recommendation[32] used in the model increases the $CH_3OOH$ concentrations by 15–20% and goes already a long way to compensating the deterioration of model performance against aircraft campaigns (Supplementary Fig. 13 and Supplementary Table 5). Other relevant processes might be also uncertain. More work is needed to address those issues.

**A large source of methanol.** In agreement with previous studies[4,5,34,35], our simulation omitting $CH_3O_2 + OH$ underestimates $CH_3OH$ observations by a factor of ∼2 over the remote Pacific (Fig. 5). When including the reaction, methanol production from $CH_3O_2 + RO_2$ is halved, from 33 to 15 Tg per year (Supplementary Table 6) due to its near-quadratic dependence on $CH_3O_2$ levels. The $CH_3O_2 + OH$ reaction would therefore worsen the model underestimations unless methanol is produced by the reaction with a sufficient yield. The model/data mismatch is barely reduced in run B, with its low overall methanol yield (∼12%), while it nearly vanishes in run C, with an overall yield of 32%. Both the average concentration and vertical profile shape are greatly improved in this case.

Similar conclusions hold for comparisons at the most remote sites Mauna Loa and Cape Verde, which was shown to be under mostly maritime and Saharan influences[36] (Table 5, Supplementary

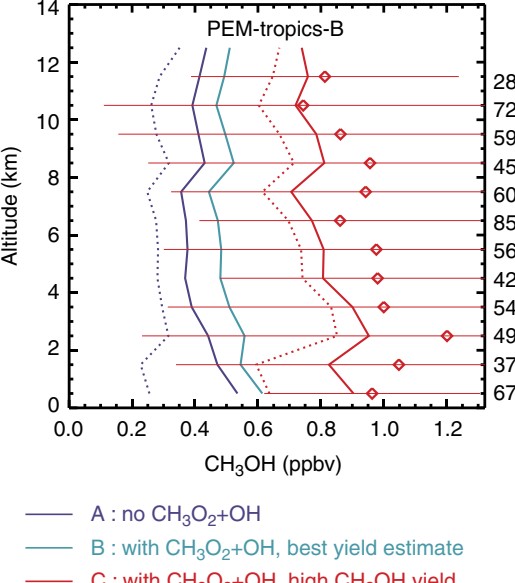

**Figure 5 | Measured and modelled methanol over the remote Tropical Pacific.** Measurements from the Pacific Exploratory Mission in the tropical Pacific (PEM-Tropics-B)[6]. Solid (dotted) lines denote simulations including (excluding) the oceanic source of methanol. The number of measurements per altitude bin is indicated on the right. The error bars represent the s.d.'s of the measurements. The nominal precision of the measurements was 25% (ref. 65).

**Table 5 | Measured CH₃OH over oceans and model biases.**

| Campaign | Area | Obs. | A | A_NO | B | C | C_NO |
|---|---|---|---|---|---|---|---|
| PEM-Tropics-B | Pacific | 934 | 0.44 | 0.30 | 0.54 | 0.87 | 0.74 |
| PEM-West-B | N.-W. Pacific | 702 | 0.83 | 0.71 | 0.99 | 1.35 | 1.26 |
| INTEX-A | N. Atlantic | 1,689 | 0.76 | 0.73 | 0.83 | 0.97 | 0.93 |
| ITCT | N. Atlantic | 991 | 0.98 | 0.95 | 0.96 | 1.15 | 1.12 |
| INTEX-B | Pacific | 1,012 | 0.60 | 0.53 | 0.67 | 0.80 | 0.73 |
| Mauna Loa | N. Pacific | 900 | 0.50 | 0.23 | 0.63 | 0.89 | 0.64 |
| Cape Verde | N. Atlantic | 768 | 0.38 | 0.18 | 0.73 | 1.06 | 0.68 |
| AMT-22 | Atlantic | 420 | 1.12 | 0.52 | 1.33 | 1.96 | 1.35 |
| INDOEX-1999 | S. Indian | 708 | 0.79 | 0.43 | 0.94 | 1.56 | 1.19 |
| Mean model bias factor* | | | 0.73 | 0.45 | 0.82 | 1.12 | 0.92 |
| Mean discrepancy factor† | | | 1.47 | 2.25 | 1.31 | 1.26 | 1.31 |

*Defined as $G = \left(\prod_{i=1}^{N} M_i/O_i\right)^{1/N}$, with $M_i/O_i$ the ratio of the averaged modelled values to the averaged observed values for campaign $i$.
†Defined as $G_{abs} = \exp[(1/N) \cdot \sum_{i=1}^{N} |\log(M_i/O_i)|]$, that is, it is the geometrically averaged ratio of the higher to the lower among the model and observed averages.
Averaged observed mixing ratios (pptv) and ratios of averaged modelled to averaged observed values for model runs defined in Table 4. See Supplementary Fig. 14 for more information on the measurements.

Fig. 15). The other campaigns listed in Table 5 were probably more impacted by continental emissions, which are likely overestimated in the model since they were derived by inverse modelling[34] while neglecting the contribution of $CH_3O_2 + OH$. Nevertheless, including the reaction improves the model performance against airborne measurement at mid-latitudes (Supplementary Fig. 16). An overestimation of oceanic emissions likely contributes also to the strong model overestimation for the Atlantic Meridional Transect cruise AMT-22 (Supplementary Fig. 17), since flux measurements by eddy covariance during that campaign[37] indicated no emission, only deposition with an average dry deposition velocity ($0.68\,\mathrm{cm\,s^{-1}}$) close to the corresponding model value ($0.63\,\mathrm{cm\,s^{-1}}$). Suppressing oceanic emissions in the model (run C_NO) leads to a much closer agreement with the latitudinal profile of AMT-22 data. That this setting leads however to large underestimations at other sites (Table 5) suggests that ocean/atmosphere exchanges are more variable than currently assumed.

## Discussion

The theoretical results presented above leave little doubt that the Criegee pathway (1) is negligible in atmospheric conditions, that is, $CH_2O_2$ is not formed in any significant amount from $CH_3O_2 + OH$ which therefore cannot be an important source of formic acid. Actually, any sizable contribution of the reaction to HCOOH formation would lead to huge overestimations of its modelled concentrations over the Southern Pacific (Supplementary Fig. 18) where median concentrations of only 19 pptv were measured in the boundary layer during spring[38], almost an order of magnitude below modelled values assuming a 40% stabilized $CH_2O_2$ yield.

The theoretical calculations further inform us that the methoxy pathway (2) is expected to dominate, whereas both methanol formation (3) and stabilization of the trioxide are viable, but likely minor. Of fundamental interest is that significant direct methanol production can occur only because the activated $CH_3OOOH$ intermediate decomposes indirectly through the $CH_3O \bullet HO_2$ complex. The share of each of the two minor channels likely does not exceed ~20% of the total reaction rate. Theory alone cannot provide precise yield estimates, due to uncertainties in key parameters, that is, precise barrier heights, average energy losses per collision, and the singlet $\leftrightarrow$ triplet ISC rate that impacts methanol production.

Fortunately, atmospheric measurements provide valuable constraints. The persistent model underestimation of $CH_3OH$ measurements over the remote Pacific[4,5,34,35] cannot be explained by ocean/atmosphere exchanges, since higher emissions would cause strong decreasing vertical gradients not seen in campaign data; on the contrary, eddy covariance measurements[37] indicate that oceanic emissions might be very low. Another remote source of methanol, the photochemical production due to $CH_3O_2 + RO_2$ reactions, could be underestimated. But since it is largely dominated by the $CH_3O_2$ self-reaction of which the methanol yield (~0.63) cannot be much underestimated, and since $CH_3O_2$ production from $CH_4 + OH$ is also well constrained, only a large production of organic peroxy radicals from non-methane organic precursors could boost this source of methanol. High observed acetaldehyde over oceans[6,36,39] suggested indeed the existence of unknown sources of $CH_3CHO$ or of its precursors. However, besides the noted inconsistency[39] of those observations with measured PAN:NOx ratios, the good agreement of modelled $CH_2O$ with observations over oceans (Supplementary Fig. 12) shows that those potential sources cannot weigh heavily on $CH_2O$ and $CH_3O_2$ production.

The only viable candidate for the missing remote source of methanol is therefore $CH_3O_2 + OH$, with an overall methanol yield crudely estimated to be ~30%, consistent with theoretical estimates given their uncertainties. Given the experimental uncertainty (50%) in the total rate constant of the reaction[9], this inferred overall yield could be even higher, or possibly lower. The total photochemical source of methanol in run C (130 Tg per year) slightly exceeds the range (50–100 Tg per year) estimated by Jacob et al.[4], because of a higher net ocean sink and lower continental emissions in our simulations. The contribution of $CH_3O_2 + OH$ (115 Tg per year globally) is comparable in magnitude to the total terrestrial emission source. Further work will be needed to evaluate how this source will impact our understanding of the methanol budget. Clearly, a better understanding of ocean-atmosphere exchanges is required to refine the top–down yield estimation presented above. It is not currently possible to determine which of the direct methanol formation channel or indirect pathway through the stabilised **TRIOX** is dominant. A more direct experimental determination is obviously needed. Hopefully **TRIOX** can be measured, although loss to walls might prove challenging, and atmospheric concentrations should be very low, a few pptv at most.

Reaction with OH might be also significant for other peroxy radicals, such as those formed from biogenic terpenoids. We expect that for the much larger activated ROOOH†, conversion to a complex RO—HO₂ and eventual dissociation to $RO + HO_2$ will be substantially slower than for $CH_3OOOH$†,

such that collisional stabilisation of ROOOH$^†$ should be a major if not dominant product route. However, the global relevance for large peroxys formed from biogenic terpenoids is expected to be lower compared with $CH_3O_2 + OH$, because OH levels are generally lower in the boundary layer over forests than in the oceanic lower troposphere, and because the reaction of $HO_2$ with large peroxys is considerably faster than the reaction of $CH_3O_2$ with $HO_2$ (ref. 40). Assuming a rate constant similar to that of $CH_3O_2 + OH$, the reaction of large biogenic peroxy radicals with OH is estimated here to account for $\sim 3\%$ of their total sink over remote forests.

## Methods

**Quantum chemistry.** All the structures presented on the potential energy surface were fully optimized with DFT using the M06-2X-D3/6-311++G(3df,3pd) level of theory in Gaussian 09 version D.01 (ref. 41; see Supplementary Table 7), except where explicitly stated otherwise. The M06-2X-D3 method was benchmarked and recommended for the general main group elements including the evaluation of thermochemistry, kinetics and noncovalent interactions[42]. For the geometry optimization and the corresponding frequency calculations, a tight convergence criterion and a superfine integration grid (150,974) were adopted. Wherever appropriate, an unrestricted ansatz was used, and careful checks were made to ensure that low-spin open-shell solutions were found to the Kohn–Sham equations. The calculated vibrational frequencies have been used for evaluation of zero-point energies and RRKM rate calculations. All the transition states for the reactions involving the abstraction or transfer of a hydrogen atom were confirmed by IRC calculations, while for the reactions from the reactant complex $^1$**RC** via **TRIOX** to the product complex $^1$**PC**, attempts to generate IRC curves failed due to the low curvature of the potential energy surface, so instead, various scans were used to explore the behaviour of those two loose TSs (Supplementary Fig. 3 and Supplementary Note 1).

Various single-point calculations were carried out using CASSCF, CASPT2 and CCSD(T)-F12. The CASSCF and CASPT2 calculations were performed with Molpro 2012.1 (refs 43,44) with an active space of 18 electrons in 13 active orbitals (which corresponds to a full-valence active space, but omitting the O–H and C–H bonding orbitals and the C–H anti-bonding orbitals), with the cc-pVTZ basis set[45]. We also calculated the spin–orbit coupling constant with Molpro at CASSCF(18,13)/cc-pVTZ level of theory.

Explicitly correlated CCSD(T)-F12 energies were obtained with the ORCA 3.03 package[46], using an unrestricted HF-reference wavefunction for reactant and product radicals and the open-shell singlet intermediates, and a restricted reference otherwise. The cc-pVTZ-F12 basis set[47] and appropriate auxiliary basis sets were used. Unless mentioned otherwise, all reported energies derive from CCSD(T)-F12/cc-pVTZ-F12 calculations with ZPVE corrections from M06-2X-D3/6-311++G(3df,3pd).

**Theoretical reaction kinetics.** The statistical-kinetics rates for thermal as well as chemically activated reactions were estimated using the energies obtained at our highest level and including the ZPVE (scaled by a factor 0.97 (ref. 48)) as stated above, while partition functions or integrated sums of vibration states and vibration state densities were based on the M06-2X-D3 harmonic vibration frequencies (scaled by a factor 0.983 (ref. 48)) and rigid-rotor rotation constants. For all structures and transition states involved we adopted the harmonic oscillator approximation, commensurate with our aim of identifying the relevant reaction pathways and distinguishing between major and minor routes based on an approximate, semi-quantitative assessment of end-product yields.

As detailed in the Results section, the only thermal bimolecular reaction that needs to be considered was found to be unimportant if not negligible. Its thermal rate coefficient at 298 K was computed using conventional transition state theory[22], with tunnelling factor estimated assuming an asymmetric Eckart potential[49,50].

The majority of the relevant reaction steps are chemically activated unimolecular reactions of (i) the initial singlet reactant complex $^1$**RC**, (ii) the closed-shell singlet intermediate **TRIOX** and (iii) the resulting product complexes $^1$**PC** and $^3$**PC**. We employed RRKM theory[21,22] to estimate the micro-canonical rate coefficients for each of the steps:

$$k(E_v) = \alpha \times G^{\neq}\left(E_v - E^{\neq}\right)/(h \times N(E_v))$$

in which $\alpha$ is the reaction path degeneracy, $h$ Planck's constant, $E^{\neq}$ the TS energy, $E_v$ the vibration energy of the reacting activated intermediate, $N(E_v)$ its density of vibration states and $G^{\neq}(E_v - E^{\neq})$ the sum of accessible vibration states of the TS; both the latter were evaluated by exact count[22]. The conservation of angular momentum was accounted for in reasonable approximation by correcting the potential energies (inclusive ZPVE) of the activated reactants and transition structures by their average rotation energy relative to that of the reactant complex $^1$**RC**, adopting the quasi-diatom approximation (see Supplementary Note 5). The changes in effective potential energy[21,22] are at most $\pm 0.3$ kcal mol$^{-1}$ and the rotational effects on the chemically activated rate coefficients remain minor for all

reactions (1–20%). Tunnelling corrections for all relevant activated reactions were found negligible because of the relatively high excess vibration energies ($E_v - E^{\neq}$) and low imaginary frequencies (see Supplementary Table 1). The rate constants for the dissociation of the singlet and triplet product complexes $^1$**PC** and $^3$**PC** were calculated using variational RRKM[21,22] by locating the structure 'var' that minimizes $G^{var}(E_v - E^{var})$. The rate constant for ISC of $^1$**PC** to $^3$**PC** was calculated based on locating the minimum energy crossing point using the code developed by Harvey and Aschi[51,52]. (see Supplementary Note 2)

In the total distributable energy $E_v$, we duly include the thermal vibration energy $E_{th,v}$ that the intermediates inherit from the thermal reactants $CH_3O_2 + OH$, and for each step the rates $k(E_v)$ were averaged over the thermal distribution function $F(E_{th,v})$ of formation. As the lifetime of the initial complex $^1$**RC** is $<1$ ps, the original thermal distribution $F(E_{th,v})$ remains conserved up to the **TRIOX** intermediate, but it shifts down for $^1$**PC** and $^3$**PC** by $\sim 0.45$ kcal mol$^{-1}$ at 298 K/1,013 hPa, $\sim 0.35$ kcal mol$^{-1}$ at 285 K/750 hPa and $\sim 0.25$ kcal mol$^{-1}$ at 256 K/400 hPa, respectively, due to the average of $\sim 0.5$, 0.4 and 0.3 collisions, respectively, that **TRIOX** suffers in the atmosphere during its lifetime (see $<k_4>$ rate values in Table 2). The lifetime of $<0.3$ ps of $^1$**PC** and $^3$**PC** is too short for further collisional losses. The width of $F(E_{th,v})$, of $\sim 2$ kcal mol$^{-1}$, is much less than the average excess energy ($E_v - E^{\neq}$) for most reaction steps, and the $k(E_v)$ are generally not far from their high-energy asymptotes, such that the $F(E_{th,v})$-averaged $<k(E_v)>$ differ only 1–20% from $k(<E_v>)$. The only exception is the $^1$**PC** reaction through **TS2**, which anyway is negligibly slow compared with the competing reaction through **TS1**. As a result, the precise shape of $F(E_{th,v})$ is of little importance for the overall kinetics of the chemically activated unimolecular reactions of our system, though the average value of the thermal $<E_{th,v}>$ is of some significance. As there is no entrance barrier and the entrance transition state **TS$_{in}$** is variational (see Fig. 1), the formation distribution function $F(E_{th,v})$ cannot be evaluated in the usual way[21]. However, the average initial $<E_{th,v}>$, of $\sim 2.1$, 2.0 and 1.75 kcal mol$^{-1}$ at 298, 285 and 256 K, respectively, could be estimated with sufficient accuracy for our purpose and a reasonable $F(E_{th,v})$ could be derived accordingly as detailed in Supplementary Note 6.

Different from the effective rates of the unimolecular reactions, the (minor) fraction $f_{stab}$ of the activated **TRIOX** that becomes collisionally stabilized depends markedly on the vibration energy of the activated **TRIOX** and is therefore quite sensitive to $F(E_{th,v})$. The stabilization competes with the much faster rearrangement **TRIOX** $\rightarrow$ $^1$**PC** through **TS4** by net rate $k^n_4(E_v)$ ($\approx 0.97 \times k_4(E_v)$ due to the reverse reaction). The Lennard-Jones collision frequency of $CH_3OOOH$ with air molecules, $k_{coll} = Z_{LJ}[M]$, is estimated[53] to be about 1.2, 0.92 and $0.55 \times 10^{10}$ s$^{-1}$ at 298 K/1,013 hPa, 285 K/750 hPa and 256 K/400 hPa, respectively. The bi-exponential energy transfer model of Troe[28] was used and implemented in a quasi-stochastic approach; the average energy transferred per collision, a critical quantity for the overall $f_{stab}$, but highly uncertain, was assumed to be $<\Delta E>_{all} = -0.9$ kcal mol$^{-1}$, amounting at 298 K to an average energy lost per down-collision $\alpha \approx 1.34$ kcal mol$^{-1}$ and average energy gained per up-collision $\beta \approx 0.44$. For a given initial energy $E_{v,in}$ of **TRIOX**, the stabilization fraction was found as the repetitive product of the successive, increasing probabilities of a (new) jth collision at constant rate $k_{coll}$ in competition with **TRIOX** $\rightarrow$ $^1$**PC** rearrangement at net rate $k^n_4(E_{j-1})$ that decreases on average after each collision:

$$f_{stab}(E_{v,in}) = \prod_{0-j}\left\{k_{coll}/\left[k_{coll} + k^n_4(E_{j-1})\right]\right\}.$$

Given the initial excess energy of **TS4** of $\sim 7$ kcal mol$^{-1}$, this product converges rapidly after some 8–10 collisions for the adopted $<\Delta E>_{all}$. The overall $f_{stab}$ was finally found by integrating over the distribution function $F(E_{th,v})$. As the net **TRIOX** conversion rate $k^n_4(E_v)$ at the initial energies is relatively high, $f_{stab}$ is small and depends strongly on $<\Delta E>_{all}$ but also markedly on $k^n_4(E_v)$ itself. Obviously, at higher altitudes in the troposphere, $f_{stab}$ decreases with decreasing pressures, but this is partially offset by the pronounced, multiple effect of the simultaneous temperature decrease: through the lower $k^n_4(E_v)$, the lower average $E_{th,v}$ to be lost for stabilization, the effect on the number density of $N_2/O_2$ and the higher $\Omega^{2,2}$ collision integral. Thus, the combined effects of the decreasing pressure $P$ and temperature $T$ in the troposphere above tropical oceans result in a dependence $f_{stab} \sim P^2 \times T^{-5}$.

Overall and net reaction rates and end-product yields were obtained in a straightforward way by duly considering the fractional contribution of each competing reaction for each intermediate in the complete scheme.

**Global modelling.** IMAGES[54,55] calculates the distribution of 172 compounds at $2° \times 2.5°$ resolution, using meteorological fields from ERA-Interim analyses of the European Centre of Medium-Range Weather Forecasts (ECMWF). Simulations were made for the year 2010 with a spin-up time of 6 months starting in July 2009. Anthropogenic emissions are obtained from a global inventory (edgar.jrc.ec.europa.eu/overview.php?v = 42) overwritten by regional inventories over Europe, Asia and the U.S. Biomass burning emissions are provided by the Global Fire Emissions Database GFEDv4s (www.globalfiredata.org). Isoprene fluxes are based on the Model of Emissions of Gases and Aerosols from Nature (MEGAN)[55]. Biogenic methanol emissions (100 Tg per year globally) are obtained from an inverse modelling study[34] using IMAGES and methanol total columns from Infrared Atmospheric Sounding Interferometer. Parameterization of ocean-atmosphere methanol exchanges follows a two-layer model resulting in an

oceanic source of 39 Tg (in 2010) and an oceanic uptake represented as dry deposition (between 46 and 66 Tg per year depending on the model simulation). The isoprene degradation mechanism has been updated to account for the revised peroxy radical kinetics of the Leuven Isoprene Mechanism LIM1 (ref. 56) as well as for the chemistry of isoprene epoxides[57]. Wet scavenging is parameterized based on ECMWF cloud and precipitation fields[58]. Dry deposition follows Wesely's resistance-in-series scheme[59], with aerodynamic resistances based on Monin-Obukhov similarity theory (ECMWF, IFS Documentation—Cy40r1, Operational implementation 22 November 2013, Part IV: Physical processes, European Centre for Medium-Range Weather Forecasts, Shinfield Park, Reading, England, 2014.) using sensible heat fluxes and friction velocities from ECMWF operational analyses and quasi-laminar layer resistances dependent on gas-phase diffusivity[60]. Surface resistances are calculated depending on mesophyll, cuticular, ground and in-canopy aerodynamic resistances[59,61,62]. The surface resistances are adjusted to provide a better match of modelled dry deposition velocities with eddy-covariance estimates over a forest by Nguyen et al.[63]; in particular, the $H_2O_2$ surface resistance becomes negligible after this adjustment.

As detailed in Supplementary Note 4, the reactions of the stabilized trioxide include the thermal conversion to the product complex $^1$PC; reaction with OH; reaction with the water dimer; and reactive uptake on aqueous aerosols. Aerosol uptake is calculated[54] based on sulfate/ammonium/nitrate and carbonaceous aerosols calculated by IMAGES, and sea salt aerosols obtained from the MACC (Monitoring Atmospheric Composition & Climate) Reanalysis (apps.ecmwf.int/datasets/data/macc-reanalysis/levtype = sfc/). The water dimer concentrations are calculated using an equilibrium constant expression[64] validated with available experimental data.

**Data availability.** The authors declare that the data supporting the findings of this study are available within the article and its Supplementary Information file. Any further relevant code and data used in the paper are available from the authors upon request.

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

## Acknowledgements

This work received support from project PRODEX ACROSAT of the European Space Agency funded by the Belgian Science Policy Office (Belspo). It was also sponsored in part by Belspo under contract SD/CS/05A (project BIOSOA) in the frame of the Science for Sustainable Development programme, and by the Office of China Postdoctoral Council (No. 20140061) through the International Postdoctoral Exchange Fellowship Program. We acknowledge the free use of merged data sets from NASA tropospheric chemistry campaigns. We thank Maite Bauwens for her assistance with data management.

## Author contributions

J.P. and J.N.H. designed and supervised the theoretical calculations; Z.L., V.S.N. and J.P. performed the theoretical calculations; J.-F.M. designed the model calculations and analysed the results; J.-F.M. and T.S. performed the model calculations; all authors discussed the theoretical and modelling results; J.-F.M., J.P., Z.L. and J.H. wrote the manuscript.

## Additional information

**Competing financial interests:** The authors declare no competing financial interests.

