## [Peer review file · Nature Communications]

Reviewers' comments:

Reviewer #1 (Remarks to the Author):

This manuscript describes a hybrid study of the title reaction -- $\text{CH}_3\text{O}_2 + \text{OH}$ to assess its product distribution, concluding as the title suggests that the reaction is a significant source of methanol in the remote troposphere. The work appears to be careful and thorough and it has sufficient general interest for publication in Nature Communications. I therefore recommend publication after relatively major revisions mostly related to the presentation.

Broadly, the work is of interest because global distributions of small oxidized organic compounds, such as methanol, are one of the great integrators of tropospheric chemistry. They can provide constraints on the overall process rates, provided that we understand the canonical reaction, methane oxidation, in detail. This work shows that methane oxidation still has some surprises in store, by introducing the $\text{CH}_3\text{O}_2 + \text{OH}$ reaction as a non-trivial CH_3O_2 sink (20% on average, up to 30% in the remote tropical atmosphere). The work is also of interest because it takes the reader from detailed quantum chemistry and statistical reaction dynamics to a global model simulation; that is just plain cool and will draw general interest. Even though many kineticists have reservations about "using the atmosphere to do kinetics" the sort of constraints applied here to rank probable reactions pathways strike me as highly sensible.

My only scientific concern is that the theoretical work appears to be largely confined to 300K and 1 atm pressure, whereas the global simulations obviously span a wide temperature and pressure range. Given a 1 atm collision frequency of about 1×10^{10} Hz and a few unimolecular rate constants that may get as low as $1 \times 10^0/\text{s}$, there is a potential for CH_3OOOH stabilization. The authors do discuss this, but they fail to discuss the overall pressure and temperature dependence of the reaction pathway in any detail. I suspect that the rates and yields will be relatively insensitive to pressure, as the system is largely chemically activated, but the pressure dependence clearly does matter. Most methane in the tropics is oxidized near the surface, but not at it, and it is not immediately obvious to me how a sensible pressure dependence would affect the stabilized ROOOH yields in the model simulation. However, this is a relatively minor pathway in the title reaction, and so these uncertainties will probably not influence the overall conclusions in any substantial way. However, the pressure and temperature dependence does need to be discussed more thoroughly in the manuscript.

My major concern is with the presentation. There are relatively few readers who would follow the detailed quantum chemistry and reaction dynamics that dominate the first 3/4 of the manuscript and then also read with interest the global modeling and appreciate the implications. Fortunately for the authors, I happen to be one of those few readers. However, in my opinion the authors should really cut to the chase when they describe the computational reaction surface and rate constants and get straight to the global modeling. They should then move most of the detailed discussion of the quantum chemistry and dynamics to the methods section. This would make an already long methods section even longer, but in my opinion it would make a better paper. The authors should ask the question "what is the minimum a global modeler needs to know about the computational chemistry?" when distilling the qc results for the main text. Without this revision, the large majority of global atmospheric chemistry readers will not survive the prologue.

A short paragraph speculating about larger RO_2 might be interesting. There are areas of atmospheric chemistry where we really care about 10% and even 2% molar yields in organic oxidation. Specifically, we care about small yields of very low vapor pressure species that can participate in new-particle formation and growth. It seems to me that almost any RO_2 should make ROOOH , and that above a few carbons, this should almost all be stabilized. This implies that a decent fraction of low- NO_x HC oxidation, like 25%, may go through this meta-stable molecular product. What will happen when for example monoterpene ROOOH collide with surface area? At what point along the "auto-oxidation" sequence of larger RO_2 is the ROOOH termination likely to

occur?? I would think it might be distributed. I have been assuming that the poly-OOH products from auto oxidation are probably quite reactive in the condensed phase, but add an -OOOH moiety to this and it could get really interesting.

Minor comments:

I20 All other fluxes are given in Tg/yr, so why not this one too? For that matter, I recommend using TgC/yr.

I43 A large Criegee COULD lead to a large formic acid source. The Criegee fate and the water rate constant remain a topic of some discussion.

I52 high measured rate CONSTANT

I58-59 subject object dissonance between "its likely importance" and "this work"

I76 The relationship between the structures and locations on the PES can be fairly confusing. Not quite sure how to fix, but right now it doesn't work.

Reviewer #2 (Remarks to the Author):

The importance of the reaction between RO₂ and OH has been poorly studied in spite of suggestions for this chemistry going back as long ago as the 1970's. Müller et al. (hereafter M2016) have combined high level ab initio calculations with 3D chemical transport modelling and model-measurement analysis to address this issue.

Overall this is a well written well thought out study, but I don't feel that it is suitable for publication in Nature COMMS as it is still very speculative.

Theoretical calculations can not provide the atmospheric chemical mechanism community the vital piece of information needed here (which has so far eluded the lab community) - the yield of CH₃O vs CH₃OH from the title reaction. The key message from M2016 is that the model can be used to infer this yield. I find this very unconvincing. The strength of the model-measurement comparison is poor at best (where are the R² values? how has the bias changed?). There is effectively no statistical comparison of the model simulations with these observational data - and no mention of satellite comparisons even though the research team are experts in this area.

This is a nice theoretical study that borders on overemphasising the results. That said, I feel the authors are probably right, but this is based on my own gut feeling and not on the evidence they have produced, which must be the evidence we use to assess how this paper stands up.

Reviewer #3 (Remarks to the Author):

Summary.

The authors present a theoretical study of the CH₃O₂ + OH reaction and its products. This reaction was recently found to proceed much more rapidly than previously thought, with potentially significant implications for atmospheric composition in the unpolluted atmosphere. The reaction products have not yet been experimentally determined, but were thought likely to involve either production of 1) CH₃O + HO₂, 2) CH₂O₂ + H₂O, or 3) CH₃OH + O₂.

Based on the theoretical work here, the authors find that the reaction mainly goes via 1) to yield $\text{CH}_3\text{O} + \text{HO}_2$. This agrees with another recent theoretical study by Bian et al. The title reaction is mainly relevant in low- NO_x , pristine, regimes where CH_3O_2 would otherwise react with HO_2 to form CH_3OOH . The implication is thus that the $\text{CH}_3\text{O}_2 + \text{OH}$ reaction shifts the chemistry more towards a radical recycling mode and lessens the importance of the radical termination mode ($\text{CH}_3\text{O}_2 + \text{HO}_2$) ... though the latter is still predominant. In other words the chemistry at low NO_x behaves a bit more like the chemistry at higher NO_x levels than we thought.

Along with the dominant pathway 1), the authors also find a significant role for production of methanol from $\text{CH}_3\text{O}_2 + \text{OH}$. They also show that pathway 2), which had been proposed as a potentially major source of atmospheric formic acid, is in fact negligible.

Finally the paper presents chemical transport model simulations to evaluate their theoretical findings and assess possible implications. They examine airborne and surface observations of a range of related chemical species. In some cases (e.g., formaldehyde, CH_3OOH , H_2O_2) there is mixed or inconclusive evidence regarding the importance of the $\text{CH}_3\text{O}_2 + \text{OH}$ reaction or its probable products. In the case of methanol however, there is a large underestimate over the remote ocean that the authors argue can only be reconciled based on the $\text{CH}_3\text{O}_2 + \text{OH}$ chemistry and subsequent methanol production at the upper end of their theoretical findings. If correct, while this is not the main pathway for the $\text{CH}_3\text{O}_2 + \text{OH}$ reaction, it nonetheless provides a major revision to current understanding of the methanol budget.

Originality/interest/impact

There are several findings here that are new and of broad importance for atmospheric composition. 1) dominant production of $\text{CH}_3\text{O} + \text{HO}_2$ from $\text{CH}_3\text{O}_2 + \text{OH}$, shifting the radical chemistry of the pristine atmosphere as noted above. 2) The fact that this reaction is a negligible source of CH_2O_2 , and thus cannot explain previously noted dramatic model underestimates for atmospheric formic acid. 3) a large revision to the global methanol budget based on CH_3OH production from $\text{CH}_3\text{O}_2 + \text{OH}$.

Thus, I support publication in Nature Communications if the authors can address the comments below.

Comments.

A major finding is the inferred change to the overall methanol budget. So whether or not this paper should have the stature of a Nature Communications paper would seem to rely on how much this matters. In other words, what is the broader importance of the modified methanol budget, beyond just the changed methanol concentrations themselves? This isn't discussed in the paper.

The reported $\text{CH}_3\text{O}_2 + \text{OH}$ rate constant from Bossolasco hinges on the CH_3O_2 absorption cross section. The value they use in that work is 2-3x larger than earlier findings. Using the alternate cross-sections would result in a correspondingly lower inferred $\text{CH}_3\text{O}_2 + \text{OH}$ rate coefficient, thus reducing the chemical impacts presented in the current paper. This uncertainty isn't discussed by Muller et al. How does this uncertainty affect your conclusions? Is there reason to trust the Bossolasco cross section over the other ones?

The degraded model agreement for CH_3OOH when including the $\text{CH}_3\text{O}_2 + \text{OH}$ chemistry would seem to be a problem, since this is the known product of the alternate low- NO_x pathway $\text{CH}_3\text{O}_2 + \text{HO}_2$ that competes with $\text{CH}_3\text{O}_2 + \text{OH}$. The authors ascribe this to uncertainties in other model processes and potentially to measurement uncertainties. That may well be the case. However, from all the model-measurement comparisons we have the following take-aways:
Formaldehyde - insensitive to inclusion of this chemistry

H₂O₂ - inconclusive; agreement improves somewhat in some cases and is degraded somewhat in others.

CH₃OOH - model performance is uniformly degraded

CH₃OH - model performance is improved for the campaign shown (but see below about that)

So, disregarding the CH₃OOH comparisons while focusing on the improved agreement for CH₃OH seems a bit like cherry-picking unless we have specific reasons to suspect the measurements or other model processes for CH₃OOH. Do we? The paper needs a stronger treatment of this point.

Given the emphasis the authors place on the implications for the atmospheric methanol budget, I'm confused why they limit the model-measurement comparisons to a single campaign from 17 years ago. There have been other airborne campaigns since then that include significant coverage over the ocean and/or in the free troposphere. Do these also tend to show the same large underestimates? Including a wider selection of data here (even if in the SI) would go a long ways to strengthening your argument about a widespread missing source of atmospheric methanol that can be explained by this CH₃O₂ + OH reaction. Given how central this point is to the paper, I don't think it's appropriate to rely solely on a single flight campaign when others are available.

Along similar lines, the authors show some surface comparisons for methanol, but it's hard to interpret these one way or another with respect to the role of atmospheric production. Concentrations at the surface will be very sensitive to the particulars of how you treat surface-atmosphere exchange and PBL mixing (as you illustrate in the sensitivity runs with modified air-sea methanol fluxes). I think the airborne data are a much stronger test here, and in particular airborne data in the free troposphere where you really isolate the problem more effectively.

This doesn't go to the heart of the paper, but perhaps worth discussing briefly - what about the importance of other RO₂ + OH reactions? Clearly CH₃O₂ is the most important of the RO₂, globally at least. But could the ensemble of other RO₂ + OH reactions have implications we should think about?

Minor.

Fig S7, method statement is incorrect for formaldehyde. HPLC was not used for formaldehyde for at least some of the data shown (INTEXA-NCAR).

Fig S9, since the measurements averaged to 2 degrees are more directly relevant to the model comparisons (since the model simulation is also at ~2 degrees), suggest modifying the plot to emphasize those over the individual measurement data points (e.g. black versus grey etc.)

We thank the Reviewers for their comments and suggestions which contributed to clarify several important aspects and improve the overall quality of our manuscript. Please find below our point-by-point reply to the raised concerns and an itemised list of main changes made to the manuscript.

Reviewer #1:

This manuscript describes a hybrid study of the title reaction -- $\text{CH}_3\text{O}_2 + \text{OH}$ to assess its product distribution, concluding as the title suggests that the reaction is a significant source of methanol in the remote troposphere. The work appears to be careful and thorough and it has sufficient general interest for publication in Nature Communications. I therefore recommend publication after relatively major revisions mostly related to the presentation.

Broadly, the work is of interest because global distributions of small oxidized organic compounds, such as methanol, are one of the great integrators of tropospheric chemistry. They can provide constraints on the overall process rates, provided that we understand the canonical reaction, methane oxidation, in detail. This work shows that methane oxidation still has some surprises in store, by introducing the $\text{CH}_3\text{O}_2 + \text{OH}$ reaction as a non-trivial CH_3O_2 sink (20% on average, up to 30% in the remote tropical atmosphere). The work is also of interest because it takes the reader from detailed quantum chemistry and statistical reaction dynamics to a global model simulation; that is just plain cool and will draw general interest. Even though many kineticists have reservations about "using the atmosphere to do kinetics" the sort of constraints applied here to rank probable reactions pathways strike me as highly sensible.

My only scientific concern is that the theoretical work appears to be largely confined to 300K and 1 atm pressure, whereas the global simulations obviously span a wide temperature and pressure range. Given a 1 atm collision frequency of about $1e10$ Hz and a few unimolecular rate constants that may get as low as $1e10/s$, there is a potential for CH_3OOH stabilization. The authors do discuss this, but they fail to discuss the overall pressure and temperature dependence of the reaction pathway in any detail. I suspect that the rates and yields will be relatively insensitive to temperature, as the system is largely chemically activated, but the pressure dependence clearly does matter. Most methane in the tropics is oxidized near the surface, but not at it, and it is not immediately obvious to me how a sensible pressure dependence would affect the stabilized ROOH yields in the model simulation. However, this is a relatively minor pathway in the title reaction, and so these uncertainties will probably not influence the overall conclusions in any substantial way. However, the pressure and temperature dependence does need to be discussed more thoroughly in the manuscript.

We agree with the reviewer and have therefore calculated the various rate coefficients (revised Table 2) and product yields, in particular that of stabilized ROOH , at various relevant pressures and associated temperatures in the troposphere (additional Table 3 in the

revised ms). After the now merged and reduced theoretical Results subsection **Potential energy surface and reaction kinetics**, we have inserted a new, brief subsection **Summary: major and minor reaction products and estimated yields in the troposphere** (lines 218-242) that summarizes the computational findings regarding the expected products and specifically describes the product yields as a function of the decreasing pressure and temperature with increasing altitude in the troposphere. The pressure- and of temperature effects have been addressed also in the revised Methods section (lines 492-497), pointing out that the important pressure effect on ROOOH stabilization is moderated by the strong but opposite temperature effect.

We should add here that in the revised manuscript, we now have corrected for the effects of angular momentum conservation by considering the effective potential energies, based on estimated average rotation energies of all species involved (lines 438-443). Compared to the earlier treatment, the present, more rigorous approach results in a lower rate of the ROOOH \rightarrow ^1PC reaction and therefore in somewhat more ROOOH stabilization (see revised Supplementary Note 5).

My major concern is with the presentation. There are relatively few readers who would follow the detailed quantum chemistry and reaction dynamics that dominate the first 3/4 of the manuscript and then also read with interest the global modeling and appreciate the implications. Fortunately for the authors, I happen to be one of those few readers. However, in my opinion the authors should really cut to the chase when they describe the computational reaction surface and rate constants and get straight to the global modeling. They should then move most of the detailed discussion of the quantum chemistry and dynamics to the methods section. This would make an already long methods section even longer, but in my opinion it would make a better paper. The authors should ask the question "what is the minimum a global modeler needs to know about the computational chemistry?" when distilling the qc results for the main text. Without this revision, the large majority of global atmospheric chemistry readers will not survive the prologue.

We considered this suggestion very carefully - we understand why the referee made it, and appreciate very much his or her concern for making our paper more accessible to readers. In the end, though, we believe that we have found a different solution that should also ensure the survival of our readers, while also reflecting the cross-disciplinary nature of the overall work. Through the use of additional headings, the merging of the two sections of the (somewhat reduced) computational part and the re-phrasing of its conclusions, and the introduction of an explicit invitation to readers who wish to do so to skip the quantum-chemical and rate-theoretical part (lines 68-70), we believe that we have better 'signposted' the routes through the manuscript that different readers may elect to follow. We note that Nature Communications welcomes contributions from cross-disciplinary fields.

A short paragraph speculating about larger RO₂ might be interesting. There are areas of atmospheric chemistry where we really care about 10% and even 2% molar yields in organic

oxidation. Specifically, we care about small yields of very low vapor pressure species that can participate in new-particle formation and growth. It seems to me that almost any RO₂ should make ROOOH, and that above a few carbons, this should almost all be stabilized. This implies that a decent fraction of low-NO_x HC oxidation, like 25%, may go through this meta-stable molecular product. What will happen when for example monoterpene ROOOH collide with surface area? At what point along the "auto-oxidation" sequence of larger RO₂ is the ROOOH termination likely to occur?? I would think it might be distributed. I have been assuming that the poly-OOH products from auto oxidation are probably quite reactive in the condensed phase, but add an -OOH moiety to this and it could get really interesting.

We thank the referee for this interesting point also raised by Reviewer#3, and have added a paragraph on this issue at the end of the revised Discussion (lines 376-386). Reaction with OH might indeed be relevant for many other peroxy radicals such as those formed in the oxidation of isoprene and monoterpenes. We expect that for these much larger activated ROOOH[†], conversion to a complex RO—HO₂ and eventual dissociation into RO + HO₂ will be substantially slower than for CH₃OOH[†], such that collisional stabilization of ROOOH should be a major if not dominant product route. However, the global relevance of the reaction for larger peroxys is expected to be lower compared to CH₃O₂ + OH, first because OH levels are generally lower in the boundary layer over vegetated areas than in the oceanic lower troposphere, and secondly because the reaction of HO₂ with large peroxy radicals such as those formed from isoprene and monoterpenes is considerably faster (factor of ~3.5) than the reaction of CH₃O₂ with HO₂ (Jenkin et al., 1997). Assuming a rate constant for RO₂+OH identical to that for CH₃O₂+OH, the reaction of isoprenyl hydroxyperoxy radicals with OH is estimated with our model to account for ca. 3% of the total sink of those peroxy radicals over remote forests. A similar share might be expected for other large peroxy radicals from biogenic VOCs.

Jenkin, M. E., Saunders, S. M. & Pilling, M. J. The tropospheric degradation of volatile organic compounds: a protocol for mechanism development. *Atmos. Environ.* **31**, 81-104 (1997).

Minor comments:

120 All other fluxes are given in Tg/yr, so why not this one too? For that matter, I recommend using TgC/yr.

Agreed. We prefer Tg/yr to TgC/yr since these were the units used in previous methanol budget studies.

143 A large Criegee COULD lead to a large formic acid source. The Criegee fate and the water rate constant remain a topic of some discussion.

We disagree on that point. Very recent laboratory experiments have shown conclusively that the stabilized CH₂OO Criegee Intermediate reacts very fast ($k \approx 5 \times 10^{-12} \text{ cm}^3 \text{ s}^{-1}$) and predominantly with water dimer (Chao et al., 2015; Lewis et al., 2015; Smith et al., 2015),

while all recent field and chamber studies indicate that stabilized CH_2O_2 reacts with water (dimer or monomer) to form either formic acid or hydroxymethyl hydroperoxide (HMHP) (Neeb et al., 1997; Lewis et al., 2015; Nguyen et al., 2016). For a large fraction, HMHP is converted to HCOOH by reaction with OH or possibly upon deposition on vegetation or on other surfaces (Nguyen et al., 2015). Note also that the stabilization fraction of the CH_2OO Criegee formed from $\text{CH}_3\text{O}_2+\text{OH}$ is expected to be nearly equal to or the 45-50% stabilization fraction of the Criegee formed in C_2H_4 ozonolysis, given their similar internal energy on formation.

Chao, W. *et al.* Direct kinetic measurement of the reaction of the simplest Criegee intermediate with water vapor. *Science* **347**, 751-754 (2015).

Lewis, T. R., Blitz, M. A., Heard, D. E. & Seakins, P. W. Direct evidence for a substantive reaction between the Criegee intermediate, CH_2OO , and the water vapour dimer. *Phys. Chem. Chem. Phys.* **17**, 4859-4863 (2015).

Smith, M. C. *et al.* Strong negative temperature dependence of the simplest Criegee intermediate CH_2OO reaction with water dimer. *J. Phys. Chem. Lett.* **6**, 2708-2713 (2015).

Neeb, P., Sauer, F., Horie, O. & Moortgat, G. K. Formation of hydroxymethyl hydroperoxide and formic acid in alkene ozonolysis in the presence of water vapor. *Atmos. Environ.* **31**, 1417-1423 (1997).

Nguyen, T. B. *et al.* Atmospheric fates of Criegee intermediates in the ozonolysis of isoprene. *Phys. Chem. Chem. Phys.* **18**, 10241-10254 (2016).

Nguyen, T. B. *et al.* Rapid deposition of oxidized biogenic compounds to a temperate forest. *Proc. Nat. Acad. Sci.* **112**, E392-E401 (2015).

152 high measured rate CONSTANT

Agreed. Word added.

158-59 subject object dissonance between "its likely importance" and "this work"

Agreed. Changed to "In view of the likely major importance of the title reaction..."

176 The relationship between the structures and locations on the PES can be fairly confusing. Not quite sure how to fix, but right now it doesn't work.

In Figure 1, we had to make a compromise between the size and hence clarity of the structures and their relationship to the PES location. Still, it appears to us that the structures of initial reactants and end products can be readily linked to the formulas on the Figure. The other four structures (^1RC , TRIOX , TS4 and ^1PC) are shown as close as possible to their location and acronym. But in order to avoid all confusion, we now labelled the structures ^1RC and ^1PC , and we also added a line to the Figure 1 caption, referring the reader to Supplementary Figure 2, which shows separately each structure together with either its formula or acronym used in this work.

Reviewer #2:

The importance of the reaction between RO₂ and OH has been poorly studied in spite of suggestions for this chemistry going back as long ago as the 1970's. Müller et al. (hereafter M2016) have combined high level ab initio calculations with 3D chemical transport modelling and model-measurement analysis to address this issue.

Overall this is a well written well thought out study, but I don't feel that it is suitable for publication in Nature COMMS as it is still very speculative.

Theoretical calculations cannot provide the atmospheric chemical mechanism community the vital piece of information needed here (which has so far eluded the lab community) - the yield of CH₃O vs CH₃OH from the title reaction. The key message from M2016 is that the model can be used to infer this yield. I find this very unconvincing. The strength of the model-measurement comparison is poor at best (where are the R² values? how has the bias changed?). There is effectively no statistical comparison of the model simulations with these observational data - and no mention of satellite comparisons even though the research team are experts in this area.

Although we might understand some of the reluctance of the referee, we disagree with these statements, as detailed below.

Role of theory. As acknowledged by Reviewer #3, the theoretical study does provide very important and valuable information, e.g. the fact that the CH₂O₂ pathway is negligible, and that the CH₃O pathway should be major, while also formation of stabilized CH₃OOOH is expected.

Using of model to infer yields. Like the referee, we would generally not propose to use a model to constrain yields or other kinetic parameters. This case is different because (1) the model underestimation of remote oceanic methanol is not specific to our model, but was found in previous modelling studies as well, (2) it is not specific to a single location, but is found as a widespread feature of model-data comparisons over remote Tropical oceans, and (3) as discussed in the manuscript, there appears to be no other viable explanation for the said underestimation.

Strength of model-data comparison. The statement of the reviewer is unfair since the biases are given in the Tables in the manuscript and in the Supplement. The correlation coefficient for methanol during PEM-Tropics-B is improved from about zero in runs A and B to 0.6 in run C. We did not include that information in the manuscript because the improvement is already obvious from Figure 5. In general for the other comparisons, the correlation coefficient does not change significantly; but we think that the bias is the really useful and meaningful metric here.

Use of satellite comparisons. Formaldehyde being essentially unaffected by the title reaction, and since its columns are generally too low over ocean for accurate detection, model comparisons with HCHO columns would be pointless. Methanol has been measured in the thermal infrared by TES and IASI, but oceanic retrievals have limited information content (low Degree of Freedom for Signal, DOFS) due to weak thermal contrast and relatively low methanol abundance (in comparison to source regions over land) (Razavi et al., 2011; Wells et al., 2014). For this reason, those retrievals were never used to probe methanol sources over ocean.

Razavi, A. *et al.* Global distributions of methanol and formic acid retrieved for the first time from the IASI/MetOp thermal infrared sounder. *Atmos. Chem. Phys.* **11**, 857-872 (2011).

Wells, K. C. *et al.* Quantifying global terrestrial methanol emissions using observations from the TES satellite sensor. *Atmos. Chem. Phys.* **14**, 2555-2570 (2014).

This is a nice theoretical study that borders on overemphasising the results. That said, I feel the authors are probably right, but this is based on my own gut feeling and not on the evidence they have produced, which must be the evidence we use to assess how this paper stand up.

We are of course glad that the referee feels we're actually right, even based on gut feeling instead of on extensive comparisons with observations and sensitivity calculations.

Reviewer #3:

Summary.

The authors present a theoretical study of the $\text{CH}_3\text{O}_2 + \text{OH}$ reaction and its products. This reaction was recently found to proceed much more rapidly than previously thought, with potentially significant implications for atmospheric composition in the unpolluted atmosphere. The reaction products have not yet been experimentally determined, but were thought likely to involve either production of 1) $\text{CH}_3\text{O} + \text{HO}_2$, 2) $\text{CH}_2\text{O}_2 + \text{H}_2\text{O}$, or 3) $\text{CH}_3\text{OH} + \text{O}_2$.

Based on the theoretical work here, the authors find that the reaction mainly goes via 1) to yield $\text{CH}_3\text{O} + \text{HO}_2$. This agrees with another recent theoretical study by Bian et al. The title reaction is mainly relevant in low- NO_x , pristine, regimes where CH_3O_2 would otherwise react with HO_2 to form CH_3OOH . The implication is thus that the $\text{CH}_3\text{O}_2 + \text{OH}$ reaction shifts the chemistry more towards a radical recycling mode and lessens the importance of the radical termination mode ($\text{CH}_3\text{O}_2 + \text{HO}_2$) ... though the latter is still predominant. In other words the chemistry at low NO_x behaves a bit more like the chemistry at higher NO_x levels than we thought.

Along with the dominant pathway 1), the authors also find a significant role for production of methanol from $\text{CH}_3\text{O}_2 + \text{OH}$. They also show that pathway 2), which had been proposed as a potentially major source of atmospheric formic acid, is in fact negligible.

Finally the paper presents chemical transport model simulations to evaluate their theoretical findings and assess possible implications. They examine airborne and surface observations of a range of related chemical species. In some cases (e.g., formaldehyde, CH_3OOH , H_2O_2) there is mixed or inconclusive evidence regarding the importance of the $\text{CH}_3\text{O}_2 + \text{OH}$ reaction or its probable products. In the case of methanol however, there is a large underestimate over the remote ocean that the authors argue can only be reconciled based on the $\text{CH}_3\text{O}_2 + \text{OH}$ chemistry and subsequent methanol production at the upper end of their theoretical findings. If correct, while this is not the main pathway for the $\text{CH}_3\text{O}_2 + \text{OH}$ reaction, it nonetheless provides a major revision to current understanding of the methanol budget.

Originality/interest/impact

There are several findings here that are new and of broad importance for atmospheric composition. 1) dominant production of $\text{CH}_3\text{O} + \text{HO}_2$ from $\text{CH}_3\text{O}_2 + \text{OH}$, shifting the radical chemistry of the pristine atmosphere as noted above. 2) The fact that this reaction is a negligible source of CH_2O_2 , and thus cannot explain previously noted dramatic model underestimates for atmospheric formic acid. 3) a large revision to the global methanol budget based on CH_3OH production from $\text{CH}_3\text{O}_2 + \text{OH}$.

Thus, I support publication in Nature Communications if the authors can address the comments below.

Comments.

A major finding is the inferred change to the overall methanol budget. So whether or not this paper should have the stature of a Nature Communications paper would seem to rely on how much this matters. In other words, what is the broader importance of the modified methanol budget, beyond just the changed methanol concentrations themselves? This isn't discussed in the paper.

It is difficult to separate the impact of the additional methanol production from the overall impact of the $\text{CH}_3\text{O}_2 + \text{OH}$ reaction. The most important effect is the increased global chemical lifetime of methane, by 0.3 years when adopting a high methanol yield (run C). This is now more clearly mentioned in the manuscript (Results section, lines 263-264). The reaction has also an impact on tropospheric ozone, which is now also briefly discussed in the same section (lines 265-267). In run C, surface ozone is decreased by up to 6% over remote tropical oceans, and by 1-2% over Western Europe, North America and East Asia during the summer (new Supplementary Figure 7).

The reported $\text{CH}_3\text{O}_2 + \text{OH}$ rate constant from Bossolasco hinges on the CH_3O_2 absorption cross section. The value they use in that work is 2-3x larger than earlier findings. Using the alternate cross-sections would result in a correspondingly lower inferred $\text{CH}_3\text{O}_2 + \text{OH}$ rate coefficient, thus reducing the chemical impacts presented in the current paper. This uncertainty isn't discussed by Muller et al. How does this uncertainty affect your conclusions? Is there reason to trust the Bossolasco cross section over the other ones?

The reported rate constant for $\text{CH}_3\text{O}_2 + \text{OH}$ is based on CH_3O_2 absorption cross sections by Farago et al. (2013) which are indeed 2-3 times higher than two previous determinations. The study by Farago et al. (2013) investigated those differences in detail and showed quantitatively and convincingly that the previously reported measurements were too low (by factors of 2-3) because of overlooking other CH_3O_2 sinks besides the self-reaction, namely, the fast $\text{CH}_3 + \text{CH}_3\text{O}_2$ reaction and diffusion out of the probed region. Therefore, yes, we have reason to trust the cross sections used by Bossolasco et al. over those previous ones.

Faragó, E. P., Viskolcz, B., Schoemaker, C. & Fittschen, C. Absorption spectrum and absolute cross sections of CH_3O_2 radicals and CH_3I molecules in the wavelength range 7473-7497 cm^{-1} . *J. Phys. Chem. A* **117**, 12802-12811 (2013).

The degraded model agreement for CH_3OOH when including the $\text{CH}_3\text{O}_2 + \text{OH}$ chemistry would seem to be a problem, since this is the known product of the alternate low- NO_x pathway $\text{CH}_3\text{O}_2 + \text{HO}_2$ that competes with $\text{CH}_3\text{O}_2 + \text{OH}$. The authors ascribe this to uncertainties in other model processes and potentially to measurement uncertainties. That may well be the case. However, from all the model-measurement comparisons we have the following take-aways:

Formaldehyde - insensitive to inclusion of this chemistry

H₂O₂ - inconclusive; agreement improves somewhat in some cases and is degraded somewhat in others.

CH₃OOH - model performance is uniformly degraded

CH₃OH - model performance is improved for the campaign shown (but see below about that)

So, disregarding the CH₃OOH comparisons while focusing on the improved agreement for CH₃OH seems a bit like cherry-picking unless we have specific reasons to suspect the measurements or other model processes for CH₃OOH. Do we? The paper needs a stronger treatment of this point.

Let's be fair. For H₂O₂, as indicated in the manuscript, the average bias is decreased when including the reaction; for CH₃OOH, the model performance is improved in the case of the ship campaign. More importantly, those two compounds are not direct reaction products, and their concentrations are only marginally influenced by the product distribution, which is the main subject of this article. For CH₃OOH, the deterioration of model performance at aircraft campaigns is a consequence of the reaction but the impact of the product distribution is negligible. We are not "cherry-picking". We showed those comparisons for completeness and could have done without since they do not (in)validate the theoretical findings. As explained in the paper, the worst model performance (INTEX-B) is found even in absence of the CH₃O₂+OH reaction (consistent with a previous modelling study) indicating that unknown processes affect the measurement or model ability to quantify this compound. Explaining the reasons for that would be completely beyond the scope of the present study.

Given the emphasis the authors place on the implications for the atmospheric methanol budget, I'm confused why they limit the model-measurement comparisons to a single campaign from 17 years ago. There have been other airborne campaigns since then that include significant coverage over the ocean and/or in the free troposphere. Do these also tend to show the same large underestimates? Including a wider selection of data here (even if in the SI) would go a long ways to strengthening your argument about a widespread missing source of atmospheric methanol that can be explained by this CH₃O₂ + OH reaction. Given how central this point is to the paper, I don't think it's appropriate to rely solely on a single flight campaign when others are available.

This is a good suggestion. We originally chose to show only PEM-Tropics-B because other campaigns had significantly more continental influence, which is an issue given the large uncertainties in terrestrial emissions. Furthermore, the impact of the CH₃O₂+OH reaction is lower at mid-latitudes compared to tropical regions. Nevertheless, we follow the suggestion of the referee and now present additional comparisons with measurements of the INTEX-A, ITCT/ICARTT and INTEX-B campaigns (lines 319-321, Supplementary Fig. 16 and Table 5). The

large additional methanol source of run C is comforted by these comparisons. In particular for INTEX-A, the gradient between continental and oceanic mixing ratios is much improved.

Along similar lines, the authors show some surface comparisons for methanol, but it's hard to interpret these one way or another with respect to the role of atmospheric production. concentrations at the surface will be very sensitive to the particulars of how you treat surface-atmosphere exchange and PBL mixing (as you illustrate in the sensitivity runs with modified air-sea methanol fluxes). I think the airborne data are a much stronger test here, and in particular airborne data in the free troposphere where you really isolate the problem more effectively.

We disagree that surface comparisons are not useful. For example, the Mauna Loa measurements are consistent with those from PEM-Tropics-B. The comparisons of the model results with aircraft campaigns suggest that the vertical profiles are reasonably well reproduced by the model (at least when averaged over a wide area). At Cape Verde, the differences between the different simulations are larger than the vertical gradients in the boundary layer (the yearly averaged simulated methanol mixing ratio is ~760 pptv at the surface vs. ~900 pptv at 900 hPa). And as the referee points out, we have presented sensitivity calculations exploring the impact of the oceanic source. Nevertheless, we have followed the suggestion of the referee and present now additional comparisons with airborne data (cf. previous comment).

This doesn't go to the heart of the paper, but perhaps worth discussing briefly - what about the importance of other RO₂ + OH reactions? Clearly CH₃O₂ is the most important of the RO₂, globally at least. But could the ensemble of other RO₂ + OH reactions have implications we should think about?

We thank the referee for this interesting point also raised by Reviewer#3, and have added a paragraph on this issue at the end of the revised Discussion (lines 376-386). Reaction with OH might indeed be relevant for many other peroxy radicals such as those formed in the oxidation of isoprene and monoterpenes. We expect that for these much larger activated ROOOH[†], conversion to a complex RO—HO₂ and eventual dissociation into RO + HO₂ will be substantially slower than for CH₃OOOH[†], such that collisional stabilization of ROOOH should be a major if not dominant product route. However, the global relevance of the reaction for larger peroxys is expected to be lower compared to CH₃O₂ + OH, first because OH levels are generally lower in the boundary layer over vegetated areas than in the oceanic lower troposphere, and secondly because the reaction of HO₂ with large peroxy radicals such as those formed from isoprene and monoterpenes is considerably faster (factor of ~3.5) than the reaction of CH₃O₂ with HO₂ (Jenkin et al., 1997). Assuming a rate constant for RO₂+OH identical to that for CH₃O₂+OH, the reaction of isoprenyl hydroxyperoxy radicals with OH is estimated with our model to account for ca. 3% of the total sink of those peroxy radicals over remote forests. A similar share might be expected for other large peroxy radicals from biogenic VOCs.

Jenkin, M. E., Saunders, S. M. & Pilling, M. J. The tropospheric degradation of volatile organic compounds: a protocol for mechanism development. *Atmos. Environ.* **31**, 81-104 (1997).

Minor.

Fig S7, method statement is incorrect for formaldehyde. HPLC was not used for formaldehyde for at least some of the data shown (INTEXA-NCAR).

This is correct. We modified the legend by adding the sentence “For formaldehyde, HPLC was used during PEM-Tropics-B (accuracy 15%), and Tunable Diode Laser absorption spectrometry was used by the NCAR team (accuracy 10%) during INTEX-A and INTEX-B”.

Fig S9, since the measurements averaged to 2 degrees are more directly relevant to the model comparisons (since the model simulation is also at ~2 degrees), suggest modifying the plot to emphasize those over the individual measurement data points (e.g. black versus grey etc.)

Thanks for the suggestion. The figure has been improved to emphasize the averaged data.

List of major changes.

- Rewriting and reduction of the theoretical part of the Results Section, addressing a request of Reviewer 1.
- New subsection and new Table 3 in the Results section on the pressure and temperature dependence of the product yields, addressing the request of Reviewer 1. The new Supplementary Figure 4 displays the zonally averaged flux through the title reaction as a function of pressure and latitude.
- New paragraph in the Methods section discussing the impact of the pressure decrease with altitude on the CH₃OOOH stabilization fraction, moderated by the strong but opposite effect of the decreasing temperature.
- A more rigorous treatment of the effects of conservation of angular momentum on the chemically activated rates in the theoretical Results section, as detailed in the revised Supplementary Note 5.
- All global model runs with the title reaction turned on were remade with the newly parameterized pressure and temperature dependence of the yields. The text, plots and Tables were adapted accordingly.
- New Supplementary Figure 7 displaying the impact of the reaction on surface ozone levels, to answer the first comment of Reviewer 3.
- New comparisons with aircraft measurements from INTEX-A, INTEX-B and ITCT/ICARTT campaigns (Supplementary Figure 15 and Table 5), addressing a major comment of Reviewer 3.
- Discussion of the potential importance of RO₂+OH for other peroxy radicals (Discussion section), as required by Reviewers 1 and 3.

Reviewers' comments:

Reviewer #1 (Remarks to the Author):

In my opinion the authors have successfully addressed the reviewer concerns. The "roadmap" approach to my presentation suggestions works well, and I am satisfied that they have now appropriately addressed the pressure dependence of the title reaction (though as far as I can tell the actual model simulations are identical to the original simulations). I do believe that the combination of computational chemistry and global modeling used here is valid and appropriate; there are numerous ways we can test hypotheses that go beyond the traditional "do kinetics in the laboratory (or maybe in silico) and then implement those findings in a chemical transport model to test against observations." In my opinion the authors present a compelling case.

Reviewer #2 (Remarks to the Author):

The authors have responded well to many of the comments raised by the reviewers.

The main concerns of the reviewers were:

1) Presentation of the paper.

The authors have made some attempts to address these comments (particularly from reviewer #1) but I don't feel they have done enough to make the paper as accessible to the atmospheric chemistry community as they could. This is still very much a great piece of theoretical work with some atmospheric implications - but is presented as a break through in atmospheric chemistry but with such complexity that the average atmospheric chemist is left puzzled as to whether or not it is. Indeed, the study cites at least three other studies that have already addressed the present issue in the past (Archibald et al., 2009; Fittschen et al., 2014; Khan et al., 2014). What is new here compared to those studies is the theoretical calculations and of course that is what must be stressed. But at the moment the paper is still not balanced for the readership. In my opinion.

2) The comparison to observations and conclusions of the modelling.

The reviewers were unanimous in their reservations of the comparison of the modelling work with the cited observations. In particular reviewer #2 was concerned about the lack of rigorous statistical analysis. The authors have addressed this area of weakness by including comparison to more aircraft campaigns. More of the same. This is very much far from the state of the art in model-observation comparisons. This is the sort of stuff that I would expect to see in a first study of the title reaction. But this is the third paper on the subject. The Khan study (2014) included some poor comparison to observations and these are not much better.

Overall, I'm not convinced that the changes to the manuscript justify publication.

Reviewer #3 (Remarks to the Author):

I concur with the authors' revisions and feel they have done a nice job of addressing the issues raised by the reviewers. My sole remaining concern has to do with the consistent degradation of the CH₃OOH simulation compared to aircraft data. I don't see this as a showstopper but believe it is too glossed over in the manuscript at present.

I agree with the authors' point in their rebuttal that 1) CH₃OOH is not a direct reaction product of CH₃O₂ + OH, and 2) that the degradation of model performance for this species is not a function of the product distribution of that reaction, which indeed is the focus of this paper. Rather, it is a function of the recently-reported CH₃O₂ + OH rate that is employed. Since CH₃OOH is the product that one otherwise gets from CH₃O₂ under low-NO_x conditions in the absence of CH₃O₂ + OH, it seems the most sensitive and direct indicator of the CH₃O₂ + OH chemistry. In other words, does the model degradation for CH₃OOH call into question the fast CH₃O₂ + OH rate? While the

theoretical treatment in this paper does not bear on the rate coefficient, if the rate were much slower it would certainly lessen the importance of the overall chemistry for CH₃OH, which is in fact a main focus of the paper. The model improvement for CH₃OH relies on both the fast rate and the product distribution. It seems we are meant to weight the improvement for CH₃OH more heavily than the degradation for CH₃OOH in our evaluation of the findings. Perhaps there are indeed legitimate reasons to do so that the authors can point to. But I feel this needs to be articulated more explicitly in the manuscript. Otherwise the reader will be left wondering if this is all overblown since the good airborne CH₃OOH agreement in the prior simulation compared to the poorer agreement in the updated simulation seems to argue against the CH₃O₂ + OH rate being all that fast.

Please find below our point-by-point reply to the remaining concerns of the reviewers, as well as an itemised list of changes made to the manuscript.

Reviewer #1:

In my opinion the authors have successfully addressed the reviewer concerns. The "roadmap" approach to my presentation suggestions works well, and I am satisfied that they have now appropriately addressed the pressure dependence of the title reaction (though as far as I can tell the actual model simulations are identical to the original simulations). I do believe that the combination of computational chemistry and global modeling used here is valid and appropriate; there are numerous ways we can test hypotheses that go beyond the traditional "do kinetics in the laboratory (or maybe in silico) and then implement those findings in a chemical transport model to test against observations." In my opinion the authors present a compelling case.

Reviewer #2:

The authors have responded well to many of the comments raised by the reviewers.

The main concerns of the reviewers were:

1) Presentation of the paper.

The authors have made some attempts to address these comments (particularly from reviewer #1) but I don't feel they have done enough to make the paper as accessible to the atmospheric chemistry community as they could. This is still very much a great piece of theoretical work with some atmospheric implications - but is presented as a break through in atmospheric chemistry but with such complexity that the average atmospheric chemist is left puzzled as to whether or not it is. Indeed, the study cites at least three other studies that have already addressed the present issue in the past (Archibald et al., 2009; Fittschen et al., 2014; Khan et al., 2014). What is new here compared to those studies is the theoretical calculations and of course that is what must be stressed. But at the moment the paper is still not balanced for the readership. In my opinion.

The paper presentation was only a concern for Reviewer #1, who is now perfectly happy with the changes we made to address this aspect (see above). The reader is free to read or not the quantum-chemical and theoretical kinetics part, with no consequence for understanding the atmospheric implications.

It is completely untrue that the only real novelty of our study is the theoretical calculations. None among the three previous studies (Archibald et al., 2009; Fittschen et al., 2014; Khan et al., 2014) seriously addressed the present issue, nor did they make any recommendation for

the product yields based on atmospheric observations. Archibald et al. (2009) and Fittschen et al. (2014) presented only box model calculations, without any comparison with methanol observations. The global modelling study of Khan et al. (2014) did not show any comparison with observations illustrating the impact of the $\text{CH}_3\text{O}_2+\text{OH}$ reaction.

Note also that the last point of Reviewer #2 – that the theoretical calculations should be stressed, more than the atmospheric implications – is in total contradiction with the original comment of Reviewer #1, who felt that the theoretical calculations were given too much importance in the Results section.

2) The comparison to observations and conclusions of the modelling.

The reviewers were unanimous in their reservations of the comparison of the modelling work with the cited observations. In particular reviewer #2 was concerned about the lack of rigorous statistical analysis. The authors have addressed this area of weakness by including comparison to more aircraft campaigns. More of the same. This is very much far from the state of the art in model-observation comparisons. This is the sort of stuff that I would expect to see in a first study of the title reaction. But this is the third paper on the subject. The Khan study (2014) included some poor comparison to observations and these are not much better.

Overall, I'm not convinced that the changes to the manuscript justify publication.

The reviewers were NOT unanimous in their reservations of the comparisons with observations. Reviewer #1 said nothing about this aspect except that he/she finds our case “compelling”. Reviewer #3 did ask for additional comparisons (which we provided) but did not question the statistical rigor of our analysis. Reviewer #2 asked for more statistics such as biases, etc. As discussed in our previous answers to the referee, the biases were already given in the article. Nevertheless, we have chosen now to include additional diagnostics in Table 5 and in Supplementary Tables 4 and 5. We provide now the geometrically averaged ratio of model to observed averaged values across all campaigns, which is a good measure of the overall bias. We also provide a mean discrepancy factor (geometric average of the higher to the lower among the modelled and observed averages across all campaigns). Both metrics indicate, as expected, that the simulations C and C_NO provide the best match with the observations for methanol (also H_2O_2).

As mentioned above, the Khan et al. (2014) study presented model comparisons with PEM-Tropics-B, but they did not investigate the impact of $\text{CH}_3\text{O}_2+\text{OH}$ on the model performance in those comparisons.

Reviewer #3:

I concur with the authors' revisions and feel they have done a nice job of addressing the issues raised by the reviewers. My sole remaining concern has to do with the consistent degradation of the CH₃OOH simulation compared to aircraft data. I don't see this as a showstopper but believe it is too glossed over in the manuscript at present.

I agree with the authors' point in their rebuttal that 1) CH₃OOH is not a direct reaction product of CH₃O₂ + OH, and 2) that the degradation of model performance for this species is not a function of the product distribution of that reaction, which indeed is the focus of this paper. Rather, it is a function of the recently-reported CH₃O₂ + OH rate that is employed. Since CH₃OOH is the product that one otherwise gets from CH₃O₂ under low-NO_x conditions in the absence of CH₃O₂ + OH, it seems the most sensitive and direct indicator of the CH₃O₂ + OH chemistry. In other words, does the model degradation for CH₃OOH call into question the fast CH₃O₂ + OH rate? While the theoretical treatment in this paper does not bear on the rate coefficient, if the rate were much slower it would certainly lessen the importance of the overall chemistry for CH₃OH, which is in fact a main focus of the paper. The model improvement for CH₃OH relies on both the fast rate and the product distribution. It seems we are meant to weight the improvement for CH₃OH more heavily than the degradation for CH₃OOH in our evaluation of the findings. Perhaps there are indeed legitimate reasons to do so that the authors can point to. But I feel this needs to be articulated more explicitly in the manuscript. Otherwise the reader will be left wondering if this is all overblown since the good airborne CH₃OOH agreement in the prior simulation compared to the poorer agreement in the updated simulation seems to argue against the CH₃O₂ + OH rate being all that fast.

We believe that there are indeed very legitimate reasons for stressing the CH₃OH improvement much more heavily than the degradation for CH₃OOH. First of all, the changes in methanol concentrations due to the CH₃O₂+OH reaction reach a factor of 2 over remote Tropical oceans, whereas the changes in CH₃OOH reach only about 25%. And secondly, the model uncertainties (and also possibly the measurement uncertainties) for CH₃OOH are very likely at least 25%.

To illustrate this last point, we performed an additional simulation (denoted C_VR) identical to run C, but using a lower rate constant for the reaction of CH₃OOH with OH, still well within its estimated uncertainty range: simulation C_VR adopts the expression of the rate constant measurement by Vaghjani and Ravishankara (1989), equal to $5.54 \times 10^{-12} \text{ molec.}^{-1} \text{ cm}^3 \text{ s}^{-1}$ at 298 K, i.e. 25% lower than the JPL recommendation which was used in all our simulations. The estimated uncertainty is a factor of 1.4 according to JPL. As can be seen from the updated Figure S13, the decreased rate for CH₃OOH+OH goes a long way to restoring the original model performance of run A against aircraft campaigns (see also Table S5). This is of course just an example, as other processes might be missing or not accurately represented in models (e.g. the rate coefficient for CH₃O₂+HO₂ producing CH₃OOH is uncertain by a factor 1.25), and measurements have their uncertainties as well.

Nevertheless, we agree with the Reviewer that the total rate constant of the reaction is uncertain and could be lower (or higher) than the experimental value adopted in the calculations. We have therefore added the following sentence in the Discussion (lines 373-375): “Given the experimental uncertainty (50%) in the total rate constant of the reaction (Bossolasco et al. 2014), this inferred yield could be even higher, or possibly lower”.

List of changes.

- Updated Table 5 and Supplementary Tables 4 and 5 which now providing additional statistics (geometrically averaged ratios) for the model comparisons against measurements.
- An additional simulation has been conducted, denoted C_VR in Table 4. It is identical to simulation C, but with a lower rate constant $k(\text{CH}_3\text{OOH}+\text{OH})$ which is however well within its estimated uncertainty range.
- Updated Supplementary Table 5 and Supplementary Figures 11 and 13, incorporating the results of additional simulation C_VR.
- The discussion of the model performance regarding CH₃OOH has been updated as follows (lines 306-312): “(...) measurement issues and/or model uncertainties likely cause the discrepancies. For example, the estimated uncertainty in the rate constant of the CH₃OOH+OH reaction is a factor of 1.4 (Burkholder et al., 2015). Adopting a rate constant measurement (Vaghjiani and Ravishankara, 1989) about 25% lower than the current JPL (Jet Propulsion Laboratory) recommendation (Burkholder et al., 2015) used in the model increases the CH₃OOH concentrations by 15-20% and goes already a long way to compensating the deterioration of model performance against aircraft campaigns (Supplementary Fig. 13 and Supplementary Table 5). Other relevant processes might be also uncertain.”
- Sentence added in the Discussion (lines 373-375): “Given the experimental uncertainty (50%) in the total rate constant of the reaction (Bossolasco et al., 2014), this inferred yield could be even higher, or possibly lower”.

REVIEWERS' COMMENTS:

Reviewer #3 (Remarks to the Author):

For my part I am satisfied with the authors' response and revised manuscript, and have no further edits or comments.